# Caspase-8 mediates inflammation and disease in rodent malaria

Larissa M. N. Pereira[1,2,3,7], Patrícia A. Assis [3,7], Natalia M. de Araújo[1,2], Danielle F. Durso[3], Caroline Junqueira[1], Marco Antônio Ataíde[1,2], Dhelio B. Pereira [4], Egil Lien[3], Katherine A. Fitzgerald[3], Dario S. Zamboni [5], Douglas T. Golenbock [1,3] & Ricardo T. Gazzinelli [1,2,3,6✉]

Earlier studies indicate that either the canonical or non-canonical pathways of inflammasome activation have a limited role on malaria pathogenesis. Here, we report that caspase-8 is a central mediator of systemic inflammation, septic shock in the *Plasmodium chabaudi*-infected mice and the *P. berghei*-induced experimental cerebral malaria (ECM). Importantly, our results indicate that the combined deficiencies of caspases-8/1/11 or caspase-8/gasdermin-D (GSDM-D) renders mice impaired to produce both TNFα and IL-1β and highly resistant to lethality in these models, disclosing a complementary, but independent role of caspase-8 and caspases-1/11/GSDM-D in the pathogenesis of malaria. Further, we find that monocytes from malaria patients express active caspases-1, -4 and -8 suggesting that these inflammatory caspases may also play a role in the pathogenesis of human disease.

[1] Instituto Rene Rachou, FIOCRUZ-MG, Belo Horizonte, MG 30190-002, Brazil. [2] Departamento de Bioquímica e Imunologia, ICB, Universidade Federal de Minas Gerais, Belo Horizonte, MG 31270-901, Brazil. [3] Department of Medicine, University of Massachusetts Medical School, Worcester, MA 01605, USA. [4] Centro de Pesquisas em Medicina Tropical, FIOCRUZ-RO, Porto Velho, RO 76812-329, Brazil. [5] Departamento de Biologia Celular Molecular e Bioagentes Patogenicos, Faculdade de Medicina de Ribeirão Preto, Universidade de São Paulo, Ribeirão Preto, SP 14049-900, Brazil. [6] Plataforma de Medicina Translacional, Fundação Oswaldo Cruz/Faculdade de Medicina de Ribeirão Preto, Ribeirão Preto, SP 14049-900, Brazil. [7] These authors contributed equally: Larissa M. N. Pereira, Patrícia A. Assis. ✉email: ricardo.gazzinelli@umassmed.edu

Malaria is a major public health problem in >90 countries, and according to the World Health Organization affects >200 million people and kills over 400,000 children each year. The rupture of parasitized red blood cells (RBCs) release pathogen and danger-associated molecular patterns such as DNA, RNA, hemozoin and glycosylphosphatidylinositol (GPI), and uric acid[1] that activate Toll-like receptors (TLRs)[2,3], Nod-like receptors (NLRs)[4–7], and the cyclic GMP–AMP (cGAS) synthase[8]. The activation of pattern-recognition receptors leads to inflammatory priming and secretion of pyrogenic cytokines, such as interleukin-1β (IL-1β) and tumor necrosis factor-α (TNFα) that are responsible for signs of disease, such as high fever, chills, and rigors. Malaria priming also leads to an exquisite sensitivity to secondary bacterial infection, in particular non-typhoidal salmonellosis, that often associates with severe disease[9–12]. Hence, a better understanding of the mechanisms that arm the innate immune cells to overreact during malaria is critical for the clinical management and prevention of severe disease.

IL-1β processing and release are tightly controlled. After translation, a pro-IL-1β inactive protein is cleaved by caspase-1 that is activated in an inflammasome-dependent process[13]. Inflammasomes are multiprotein complexes assembled by the oligomerization of NLRs and AIM2. The activation of caspase-1 requires an inflammasome assembly that is under the control of two signaling checkpoints. First, TLR signaling leads to enhanced transcription of inflammasome components, pro-caspase-1 and pro-IL-1β genes[14,15]. Then, a second signal triggered by NLRs or AIM2 activation promotes inflammasome assembly[16]. This second signal is induced by many stimuli, including DNA for AIM2, urate crystal, ATP, oxygen reactive, as well as microbial infections that activate NLRP3[16,17].

When challenged in vitro with hemozoin, monocytes release IL-1β dependent on inflammasome and caspase-1 activation[4,5]. Importantly, it was demonstrated that hemozoin binds to *P. falciparum* nucleic acids and traffics into the lysosomal compartment and the cytosol of host cells, where parasite DNA activates TLR9[6]. Furthermore, it was demonstrated that monocytes from febrile malaria patients express high levels of NLRP3, NLRP12, and AIM2 inflammasome specks, as well as active caspase-1[18,19]. While experimental cerebral malaria (ECM) develops in an inflammasome-independent manner, caspase-1 activation mediates hypersensitivity to endotoxin or superinfection with *Salmonella typhimurium*, and this is attenuated in *P. chabaudi*-infected NLRP3, NLRP12, or ASC knockout mice[18].

A noncanonical inflammasome pathway that regulates IL-1β production was also described[20,21]. Caspase-11 in mice and the homolog caspase-4 in humans lead to the processing of pro-IL-1β and promotes the pyroptosis by gasdermin-D (GSDM-D) cleavage, independent of inflammasome activation[22,23]. The N-terminal product of GSDM-D binds to lipid and oligomerizes in cell membranes to form pores[24]. Then, these pores promote $Ca^{2+}$ unbalance, the pyroptosis, and IL-1β release, as a consequence[25,26]. Yet another member of this family of proteases that plays an essential role in programmed cell death, caspase-8, has recently been shown to promote NF-κB activation and induce the synthesis of pro-IL-1β[27–29]. The role of Caspase-8 on IL-1β release seems to be more complex as it is also involved in GSDM-D cleavage[30]. However, their role in malaria pathogenesis has not been addressed.

Here, we demonstrate that in vivo *Plasmodium* infection is sufficient to activate caspases-11 (or caspase-4) and 8, both in *Plasmodium*-infected mice and malaria patients. Our awards indicate that while caspase-11 is dispensable, both caspase-1 and caspase-8 are important mediators of IL-1β and TNFα secretion, hypersensitivity to septic shock as well as the development of ECM.

## Results

**Expression and activation of caspase-11 in *P. chabaudi* infection**. *Plasmodium* infection induces pro-inflammatory priming leading to hypersusceptibility to septic shock. To study the importance of this pro-inflammatory priming on malaria pathogenesis, we used a low-dose Lipopolysaccharide (LPS) challenge in *Pc*-infected mice[18,31,32]. Previous studies demonstrate that in Gram-negative bacterial infection, caspase-11 is involved in caspase-1 activation upstream of inflammasome[21]. The immunoblot using splenocytes reveals that pro-caspase-11 (p43 and p38 subunits) is induced in *Pc*-infected C57BL/6 mice. The cleaved subunit of caspase-11 (p30) was found in the splenocyte supernatants. As expected, we did not find caspase-11 in samples from 129S6 (caspase-11 mutant mice), $Casp11^{-/-}$, and $Casp1/11^{-/-}$ mice (Fig. 1a). Splenocytes from $Casp1^{-/-}/11^{tg}$ mice showed regular induction and activation of caspase-11, indicating that cleavage of caspase-11 does not require caspase-1 (Fig. 1b). We also performed an immunoblot developed with an anti-caspase-1 antibody in splenocyte lysates from $Casp11^{-/-}$ mice. We found the p35 caspase-1 subunit in splenocytes from *Pc*-infected mice, indicating that caspase-1 is cleaved by a caspase-11-independent mechanism (Fig. 1c)[33].

**Expression of caspase-11 is dependent on IFNγ and NAS-TLRs.** Different studies also demonstrate that pro-caspase-11 is induced by Type I IFN as well as TLR agonists, such as LPS, Poly IC, or PAMCys3[34–37]. Priming with TLR agonists or IFNs induces pro-caspase-11 expression, but require a second challenge with LPS to activate caspase-11. In these models, LPS-induced lethality does not require Toll-like receptor 4 (TLR4)[34,35]. We found that the induction of pro-caspase-11 and cleavage of p30 caspase-11 subunit in splenocytes from *Pc*-infected was independent of TLR4 (Supplementary Fig. 1a). However, the high levels of circulating IL-1β and IL-1β produced by splenocytes from infected $Tlr4^{-/-}$ mice stimulated with LPS were very low when compared to infected C57BL/6 mice (Supplementary Fig. 1b). When we evaluated splenic CD11b⁺ cells from infected $Tlr4^{-/-}$ mice challenged with LPS, we observed lower intracellular levels of pro-IL-1β in $Tlr4^{-/-}$ than in C57BL/6 mice (Supplementary Fig. 1c). Furthermore, infected $Tlr4^{-/-}$ mice were resistant to LPS challenge (Supplementary Fig. 1d).

*Plasmodium* is a potent activator of TLR7, TLR9, and other cytosolic receptors that sense nucleic acids[2,6,31,38–40]. In addition, IFN-priming of innate immune cells is a hallmark of both human and murine malaria[32,39,40]. Hence, we investigated the levels of caspase-11 in mice lacking functional NAS-TLRs, IFNγ (*Ifng*), or IFNα/β receptor (*Ifnar1*) genes. Consistent with previous studies[35–37,41], we found that both IFNγ and Type I IFNs are key cytokine-priming differentiated monocytes for caspase-11 expression. The results presented in Supplementary Fig. 2a show that in vitro, IFNγ, and IFN Type I are sufficient to induce caspase-11 expression, and that caspase-11 p30 subunit is only detected when bone marrow-derived macrophages (BMDMs) were stimulated with iRBC. Expression of caspase-11 and its active form was not affected in total splenocytes from single *Tlr3*-, *Tlr7*-, and *Tlr9*-deficient mice infected with *Pc* (Supplementary Fig. 2b–d). Nevertheless, we found that the expression of pro-caspase-11 was highly impaired in splenocytes from *Pc*-infected $Tlr3/7/9^{-/-}$, $Ifng^{-/-}$, or $Ifnar1^{-/-}$ mice (Fig. 1d). Expression of pro-caspase-1 was not impaired, whereas cleavage and generation of active caspase-1 were affected in total splenocytes from *Pc*-infected $Ifng^{-/-}$, but not $Ifnar1^{-/-}$ mice (Fig. 1e).

**Caspase-1 has a dominant role in mediating IL-1β release in *Pc* infection**. A high proportion of circulating monocytes become

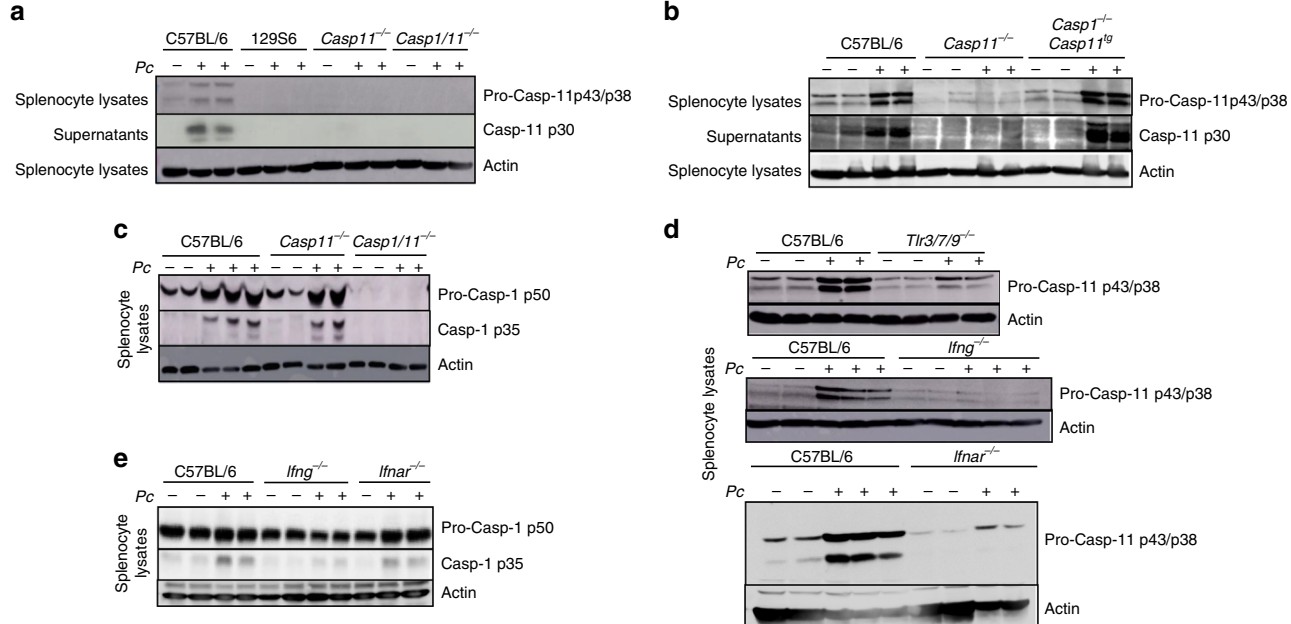

**Fig. 1 Caspase-11 activation in *P. chabaudi*-infected mice.** All results presented in this figure were obtained from mice either uninfected or at 8 days after infection with *Pc*, as indicated. **a**, **b** Splenocyte lysates and supernatants were obtained from C57BL/6, 129S6, *Casp11^{-/-}*, *Casp1/11^{-/-}*, and *Casp1^{-/-}/11^{tg}* mice, and analyzed by western blot using an anti-caspase-11 antibody. **c** Splenocyte lysates from C57BL/6, *Casp11^{-/-}*, and *Casp1/11^{-/-}* mice were analyzed by western blot using an anti-caspase-1 antibody. **d** Splenocyte lysates obtained from C57BL/6, *Tlr3/7/9^{-/-}*, *Ifng^{-/-}*, and *Ifnar^{-/-}* mice were analyzed by western blot using an anti-caspase-11 antibody. **e** Splenocyte lysates were obtained from C57BL/6, *Ifng^{-/-}*, and *Ifnar^{-/-}* mice and analyzed by western blot using an anti-caspase-1 antibody. Blots developed with an anti-β-actin were used as loading controls. All blots are representative of two to three different experiments with similar results.

activated during malaria[18,42,43]. In rodent malaria, most splenic monocytes differentiate into inflammatory monocytes (iMOs, CD11b+F4/80+CD11c−MHC II−), and monocyte-derived dendritic cells (MO-DCs, CD11b+F4/80+CD11c+MHC II+)[44] and the differentiated cells are the main source of active caspase-1 and IL-1β at 8 days post infection with *Pc*[18]. To investigate whether expression of pro-IL-1β is affected in *Casp11^{-/-}* or *Casp1/11^{-/-}* mice, we performed intracellular staining for IL-1β in iMOs and MO-DCs from spleens of control and *Pc*-infected mice, before and 2 h after LPS challenge (Supplementary Fig. 3 and Fig. 2a). We found the similar intensity of fluorescence of IL-1β in cells from either infected C57BL/6, *Casp11^{-/-}*, or *Casp1/11^{-/-}* mice. We then accessed the role of caspase-11 on IL-1β. Lower levels of IL-1β were found in culture supernatants of LPS-stimulated splenocytes from infected *Casp11^{-/-}* mice when compared to infected C57BL/6 mice. However, the IL-1β levels in splenocytes from *Casp11^{-/-}* were significantly higher than those from *Casp1^{-/-}/11^{tg}* or *Casp1/11^{-/-}* mice (Fig. 2b). In addition, when compared to infected C57BL/6 mice challenged with a low dose of LPS, lower levels of circulating IL-1β were observed in the plasma of infected *Casp1/11^{-/-}*, but not from *Casp11^{-/-}* mice (Fig. 2c). Hence, the *Casp1^{-/-}/11^{tg}* phenocopied the *Casp1/11^{-/-}* and not the *Casp11^{-/-}* mice infected with *Pc*. The plasma IL-1β levels in both control and infected mice not challenged with LPS were below the limit of detection. Altogether, our results suggest that in *Pc*-infected mice challenged with low-dose LPS, the canonical pathway of caspase-1 activation has the predominant role in IL-1β release. Consistently, deficiency of caspase-1 and not caspase-11 resulted in a small, but the significant increment of resistance to the low-dose LPS challenge in *Pc*-primed mice (Fig. 2d). Furthermore, opposite to *Ifng^{-/-}* mice[24], the *Ifnar1^{-/-}* mice that are defective on caspase-11, but not on caspase-1 expression, were still highly susceptible to the low-dose LPS challenge (Fig. 2e).

GSDM-D is cleaved by caspase-11 and caspase-1, generating the p30 polypeptide, which polymerizes and forms pores in the cell surface membrane of activated monocyte/macrophages. The GSDM-D pores allow the release of IL-1β and also lead to pyroptosis[22–24,45]. Consistent with the hypothesis of a dominant role of the canonical pathway of caspase-1 activation, *Pc*-infected *Gsdmd^{-/-}* mice phenocopied the *Casp1^{-/-}* rather than *Casp11^{-/-}* mice. Despite the severe impairment of IL-1β release (Fig. 2f), as observed in the *Casp1/11^{-/-}* and *Casp1^{-/-}* mice, the *Gsdmd^{-/-}* mice were only partially resistant to the low-dose LPS challenge (Fig. 2g).

**Caspase-8 mediates IL-1β release in *Pc*-infected mice.** The *Casp8^{-/-}* mice are not viable, but the viability is rescued in the double *Ripk3^{-/-}/Casp8^{-/-}* mice, suggesting that unregulated necroptosis is the main cause of embryonic lethality in the caspase-8-deficient mice. Therefore, we used *Ripk3^{-/-}* mice as controls. The results are shown in Fig. 3a (top panel) indicate that in vivo *Pc* infection leads to pro-caspase-8 expression (p55) and cleavage of caspase-8 (p18) in monocytes. These results were confirmed using a different anti-caspase-8 antibody that recognizes caspase-8 p43- and p18-cleaved polypeptides in CD11b+ cells Fig. 3a (bottom panel). The cleavage of caspase-8 was independent of both caspase-1 and caspase-11. In addition, our results show that caspase-8 is not necessary for cleavage of either caspase-1 or caspase-11 (Fig. 3b–d). Consistently, cleavage of caspase-8 was unimpaired in *Ifng^{-/-}* mice infected with *Pc* (Fig. 3e), which are defective on caspase-1 cleavage and caspase-11 expression.

Caspase-8 mediates TNFα-induced cell death and also promotes NF-κB activation and production of pro-inflammatory cytokines[46], including TNFα[28,29], which is an important biomarker of severe malaria[47,48]. To evaluate the importance of TNFα on caspase-8 activation, we used mice

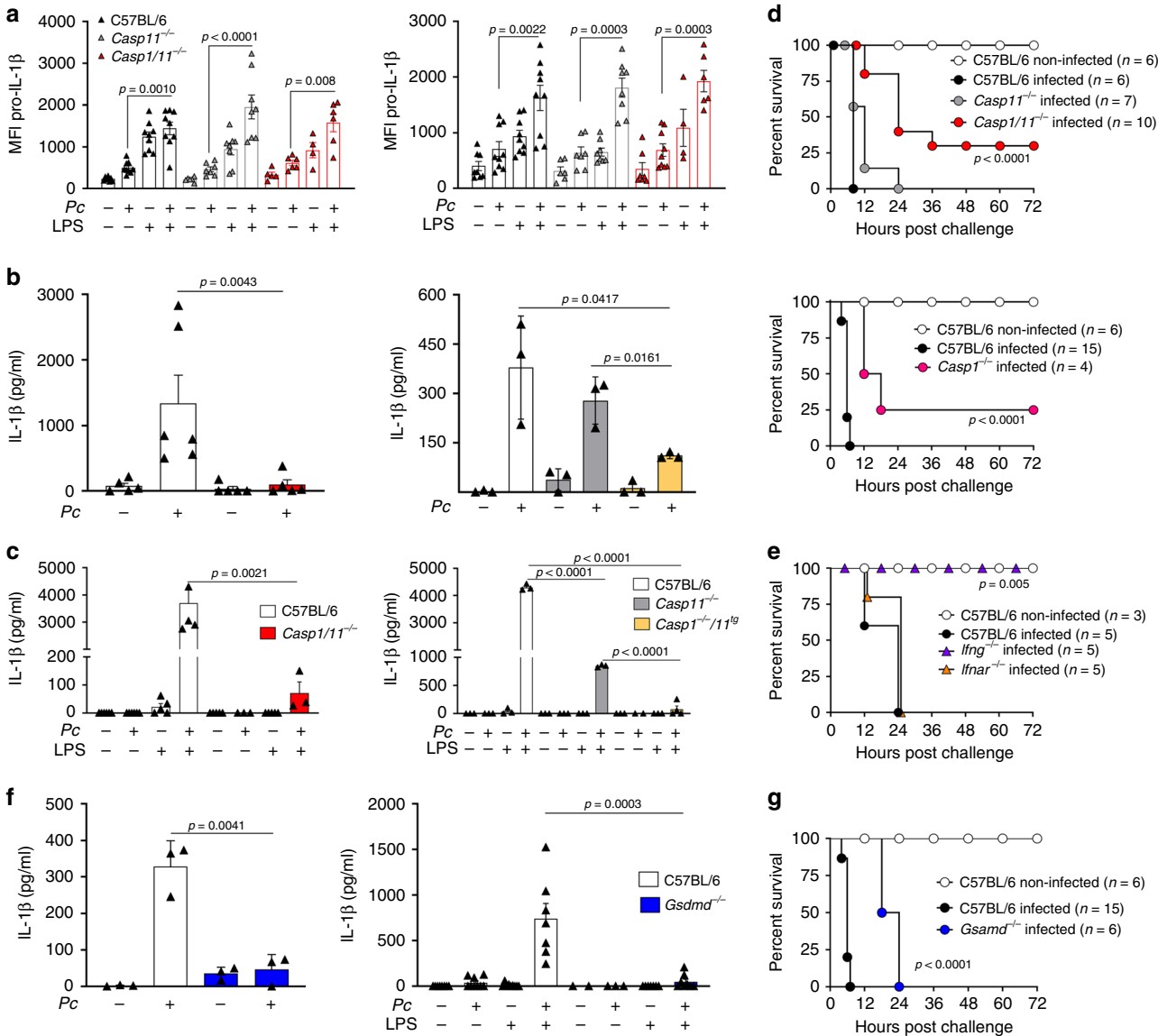

**Fig. 2 Caspase-1-dependent release of IL-1β in *Pc*-infected mice challenged with LPS. a** Splenocytes were stimulated with 1 μg/mL of LPS and used for intracellular pro-IL-1β staining by flow cytometry. The pro-IL-1β MFI quantification was assayed in live (live/dead⁻) monocytes (CD11b⁺F4/80⁺, left panel) and dendritic cells (CD11c⁺MHC II⁺, right panel). For MOs: C57BL/6 *n* = 9, *Casp1/11*⁻/⁻ (control *n* = 5, + LPS *n* = 4; infected *n* = 6), and *Casp11*⁻/⁻ (control *n* = 6, +LPS *n* = 8; infected *n* = 7, +LPS *n* = 8). For monocyte-derived dendritic cells (MO-DCs): C57BL/6 *n* = 9, *Casp1/11*⁻/⁻ (control *n* = 7, +LPS *n* = 4; infected *n* = 9, +LPS *n* = 6), and *Casp11*⁻/⁻ (control *n* = 6, +LPS *n* = 8; infected *n* = 7, + LPS *n* = 8). **b** Splenocytes were stimulated in vitro with 1 μg/ml of LPS and 24 h later IL-1β measured by ELISA in supernatants. Left panel: C57BL/6 (control *n* = 5, infected *n* = 6) and *Casp1/11*⁻/⁻ *n* = 5. Right panel: *n* = 3. **c** Mice were inoculated intravenously with 10 μg/mice of LPS, and 8 h later, plasma was collected and circulating IL-1β measured by ELISA. Left panel: C57BL/6 (non-challenged *n* = 6, LPS-challenged *n* = 5) and *Casp1/11*⁻/⁻ (non-challenged *n* = 5, LPS-challenged *n* = 3). Right panel: *n* = 3 for all groups, except *Casp1*⁻/⁻/*11*^tg (infected *n* = 2; infected +LPS *n* = 4). **d** C57BL/6, *Casp11*⁻/⁻, *Casp1/11*⁻/⁻ (top panel), as well as C57BL/6 and *Casp1*⁻/⁻ (bottom panel) were inoculated intravenously with 10 μg/mouse of LPS, and survival followed for 72 h. **e** C57BL/6, *Ifng*⁻/⁻, and *Ifnar*⁻/⁻ were inoculated intravenously with 10 μg/mouse of LPS and survival followed for 72 h. **f**, left panel: Splenocytes were stimulated in vitro with LPS (1 μg/ml) and IL-1β measured by ELISA in 24 h supernatants. For all groups, *n* = 3. **f**, right panel: C57BL/6 and *Gsdmd*⁻/⁻ mice were inoculated i.v. with 10 μg/mouse of LPS and plasma collected to measure the levels of IL-1β at 8 h post-challenge. C57BL/6 (*n* = 9, except infected +LPS *n* = 7) and *Gsdmd*⁻/⁻ (control *n* = 2 and control +LPS *n* = 7; infected *n* = 3 and infected +LPS *n* = 8). **g** C57BL/6 and *Gsdmd*⁻/⁻ mice were challenged with 10 μg/mouse of LPS and followed for survival. In **a**, **b**, **c**, and **f**, mean ± s.e.m of two-three different experiments are shown; statistical analysis by parametric or nonparametric *t* test and one-way ANOVA. Survival curves were analyzed by log-rank test.

deficient in the TNFp55 receptor gene *Tnfrsf1a* (here identified as *Tnfr*⁻/⁻). Our results show that cleavage of caspase-8 was not observed in lysates from CD11b⁺ cells derived from the spleen of *Pc*-infected *Tnfr*⁻/⁻ mice (Fig. 3f). In contrast, expression and cleavage of either of caspase-1 or caspase-11 were not affected (Fig. 3f).

The results presented in Fig. 4a show that the levels of circulating IL-1β on *Ripk3*⁻/⁻/*Casp8*⁻/⁻ or *Ripk3*⁻/⁻/*Casp8/1/11*⁻/⁻ mice are very low when compared to C57BL/6 mice infected with *Pc* and challenged with low-dose LPS. Likewise, IL-1β release by splenocytes (Fig. 4b) and pro-IL-1β expression by CD11b⁺/F4/80⁺ iMOs/MO-DC populations

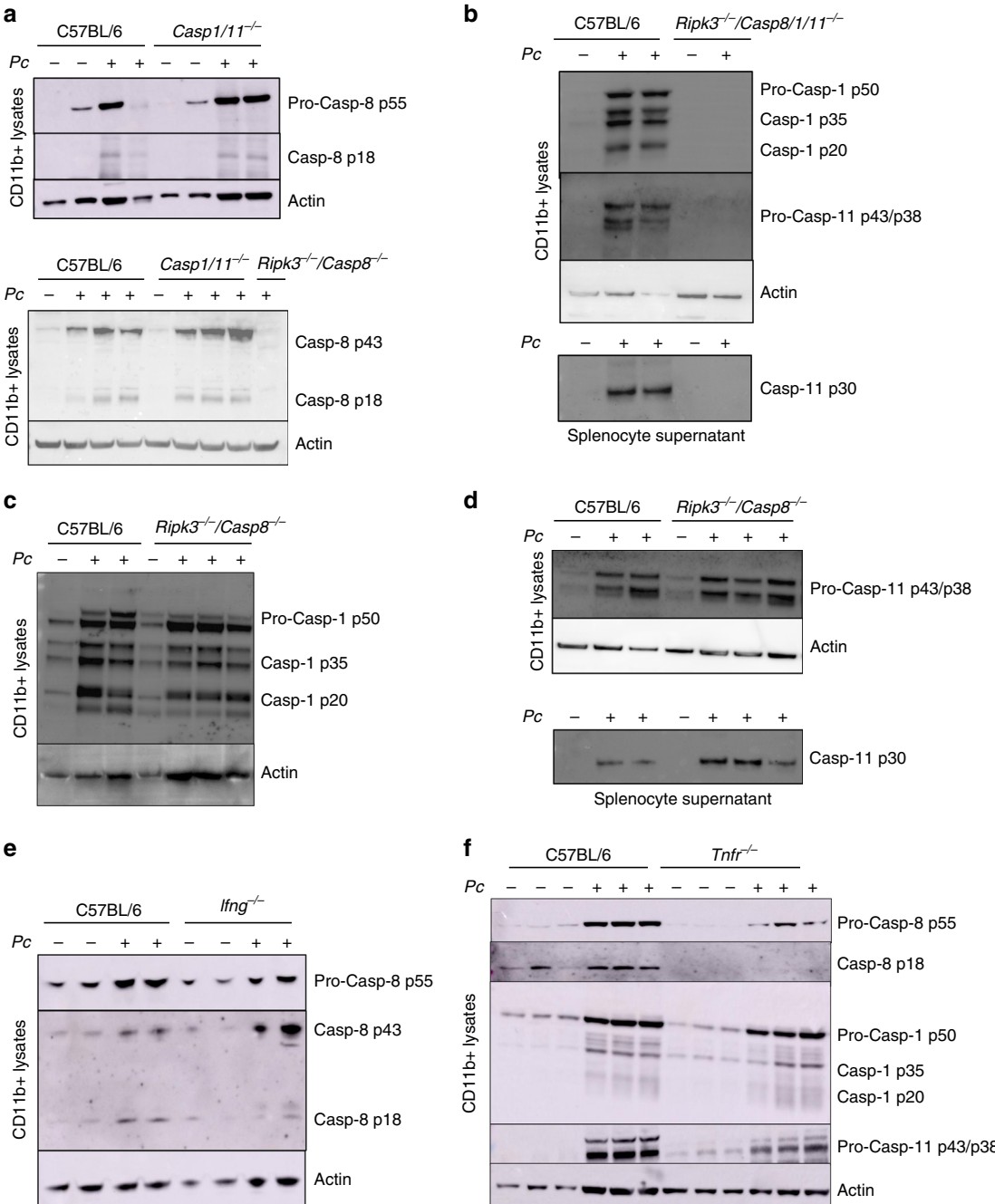

**Fig. 3 Caspase-8 activation in *Pc*-infected mice.** All results presented in this figure were obtained from mice either uninfected or at 8 days after infection with *Pc*. C57BL/6, *Casp1/11*⁻/⁻, *Ripk3*⁻/⁻/*Casp8*⁻/⁻, *Ripk3*⁻/⁻/*Casp8/1/11*⁻/⁻, *Ifng*⁻/⁻, and *Tnfr*⁻/⁻ mice were used in different experiments, as indicated. CD11b⁺ cells purified from splenocytes were lysed in RIPA buffer to prepare the lysates. Alternatively, splenocytes were spin down 400 × *g* for 5 min to prepare supernatants. Lysates from CD11b⁺ cells isolated from spleens of control and *Pc*-infected mice were analyzed by western blot using either **a** two distinct anti-caspase-8; **b** anti-caspase-1 and anti-caspase-11; **c** anti-caspase-1; **d** anti-caspase-11; and **e** anti-caspase-8 antibodies. **b**, **d** Alternatively, splenocytes supernatants were used to detect active caspase-11 (p30) by western blot using an anti-caspase-11 antibody. **f** Lysates of CD11b⁺ cells from uninfected and infected C57BL/6 and *Tnfr*⁻/⁻ mice were analyzed by western blot with anti-caspase-8, anti-caspase-1, and anti-caspase-11 antibodies. Blots revealed with anti-β-actin were used as loading controls. Blots are representative of two to three different experiments with similar results.

(Fig. 4c) from both *Pc*-infected *Ripk3*⁻/⁻/*Casp8*⁻/⁻ and *Ripk3*⁻/⁻/*Casp8/1/11*⁻/⁻ mice were impaired. As observed for IL-1β, TNFα production by splenocytes from *Pc*-infected mice was impaired in mice with various gene deficiencies (Fig. 4d). Consistently, the release of IL-1β was highly impaired in *Pc*-infected *Tnfr*⁻/⁻ mice, which are also resistant to LPS challenge (Supplementary Fig. 4a, b). Importantly, while the *Pc*-infected *Ripk3*⁻/⁻/*Casp8*⁻/⁻ mice were only partially resistant,

the infected *Ripk3*⁻/⁻/*Casp8/1/11*⁻/⁻ and *Ripk3*⁻/⁻/*Casp8/Gsdmd*⁻/⁻ mice were highly resistant to the low-dose LPS-induced lethality (Fig. 4e). The results obtained from *Ripk3* single-knockout mice were identical to C57BL6, showing that the deletion of this gene did not contribute to the phenotype observed in the double *Ripk3*⁻/⁻/*Casp8*⁻/⁻ or the quadruple *Ripk3*⁻/⁻/*Casp8/1/11*⁻/⁻-deficient mice. These results also indicate a loop of amplification that involves TNFα induction by

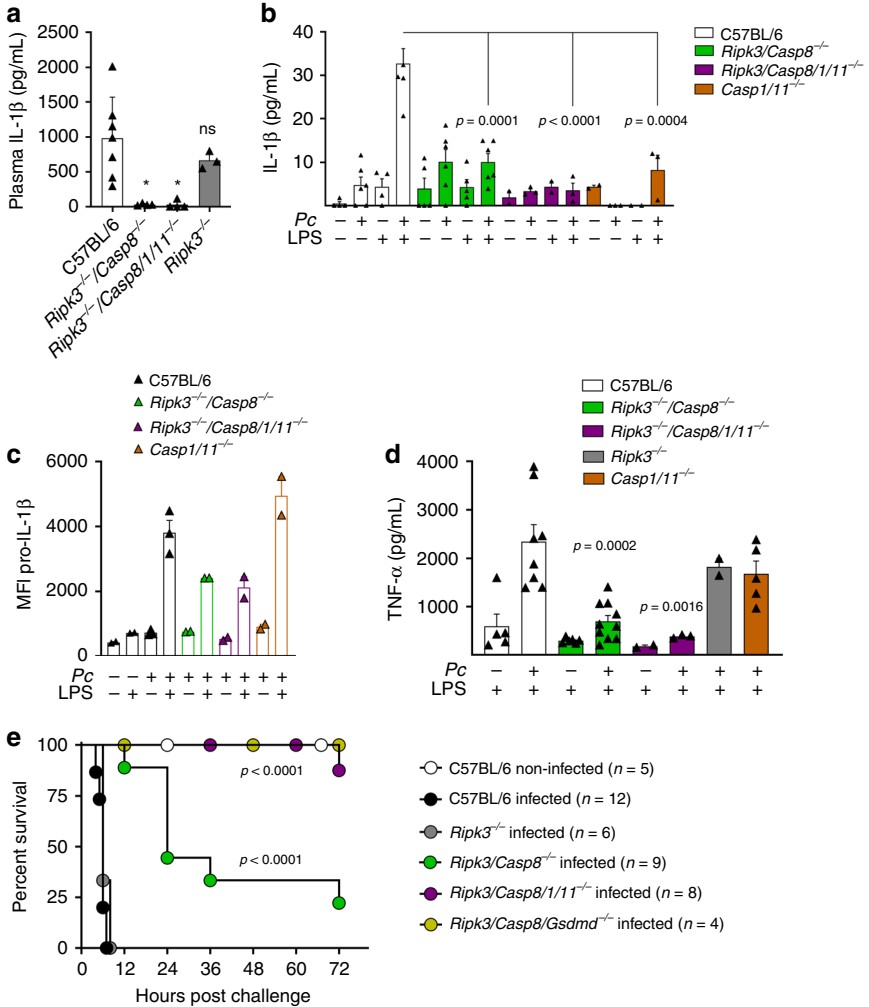

**Fig. 4 Caspase-8-mediated IL-1β release and lethality in *Pc*-infected mice challenged with LPS.** All results presented in this figure were obtained from mice either uninfected or at 8 days after infection with *Pc*. C57BL/6, *Casp1/11[−/−]*, *Ripk3[−/−]*, *Ripk3[−/−]/Casp8[−/−]*, *Ripk3[−/−]/Casp8/1/11[−/−]*, and *Ripk3[−/−]/Casp8[−/−]/Gsdmd[−/−]*. **a** Uninfected and *Pc*-infected mice were inoculated intravenously with 10 μg/mouse of LPS, and 8 h later plasma was collected to measure the levels of circulating IL-1β. C57BL/6 (n = 7), *Ripk3[−/−]/Casp8[−/−]* (n = 4), *Ripk3[−/−]/Casp8/1/11[−/−]* (n = 4) and *Ripk3[−/−]* (n = 3). *$P = 0.01$. **b** Splenocytes from uninfected controls and *Pc*-infected mice were stimulated in vitro with 1 μg/ml of LPS, and 24 h later the levels of IL-1β measured in culture supernatants by ELISA. For uninfected mice: C57BL/6 (n = 4), *Ripk3[−/−]/Casp8[−/−]* (n = 5), *Ripk3[−/−]/Casp8/1/11[−/−]*, and *Casp1/11[−/−]* (n = 2). For infected mice: C57BL/6 and *Ripk3[−/−]/Casp8[−/−]* (n = 6), *Ripk3[−/−]/Casp8/1/11[−/−]*, and *Casp1/11[−/−]* (n = 3). **c** Splenocytes from uninfected and infected mice were stimulated with 1 μg/mL of LPS, and 2 h later used for intracellular staining of pro-IL-1β. The pro-IL-1β quantification in live (live/dead−) monocytes (CD11b + F4/80 + ) was performed by flow cytometry. For all groups: n = 2, except infected C57BL/6 (n = 3). **d** Splenocytes from uninfected controls and *Pc*-infected mice were stimulated in vitro with 1 μg/ml of LPS, and 24 h later the levels of tumor necrosis factor (TNF)α measured in culture supernatants by ELISA. For uninfected mice: C57BL/6 (n = 5), *Ripk3[−/−]/Casp8[−/−]* (n = 6), *Ripk3[−/−]/Casp8/1/11[−/−]* (n = 2). For infected mice: C57BL/6 (n = 8), *Ripk3[−/−]/Casp8[−/−]* (n = 10), *Ripk3[−/−]/Casp8/1/11[−/−]* (n = 3), *Casp1/11[−/−]* (n = 5), and *Ripk3[−/−]* (n = 2). **e** Uninfected and *Pc*-infected mice were inoculated intravenously with 10 μg/mouse of LPS and survival followed for 72 h. Mean ± s.e.m of two to three different experiments are shown; statistical analysis by unpaired *t* test and one-way ANOVA. The survival curve was analyzed by log-rank test.

caspase-8, TNFα-induced caspase-8 cleavage, and consequent activation of IL-1β. Hence, our results indicate that the deleterious role of caspase-8 in the hypersensitivity to septic shock in this malaria model may also involve the induction of TNFα expression.

**A neuroinflammation gene signature in MO-DCs from *PbA*-infected mice.** In our earlier studies, we found that MO-DCs are an important source of active caspase-1, IL-1β and have a pathogenic role in ECM and acute respiratory distress syndrome in malaria (MA-ARDS) models[18,44,49]. To address the importance of inflammatory caspases in ECM, RNA was extracted from MO-DCs (CD11b+F4/80+CD11c+DC-Sign+MHC II+, purity >97%)

highly purified from spleens of control and infected mice (Supplementary Fig. 3). The MO-DCs were purified at day 5 post infection, since in this model, mice become very sick by days 6–7 post infection. The bar graph presented in Fig. 5a shows the main inflammatory pathways that were enriched in MO-DCs from *PbA*-infected mice. We found that three pathways were highly relevant to our study, i.e., maturation of dendritic cells, neuroinflammation signaling, and NFκB signaling, which is promoted by caspase-8. Importantly, the connectivity diagram shows that both TNFα and IL-1β genes were involved in all three pathways (Fig. 5b). In addition, we found that caspase-1, caspase-11, and caspase-8 genes were all upregulated in splenic MO-DCs from *PbA*-infected mice (Fig. 5c).

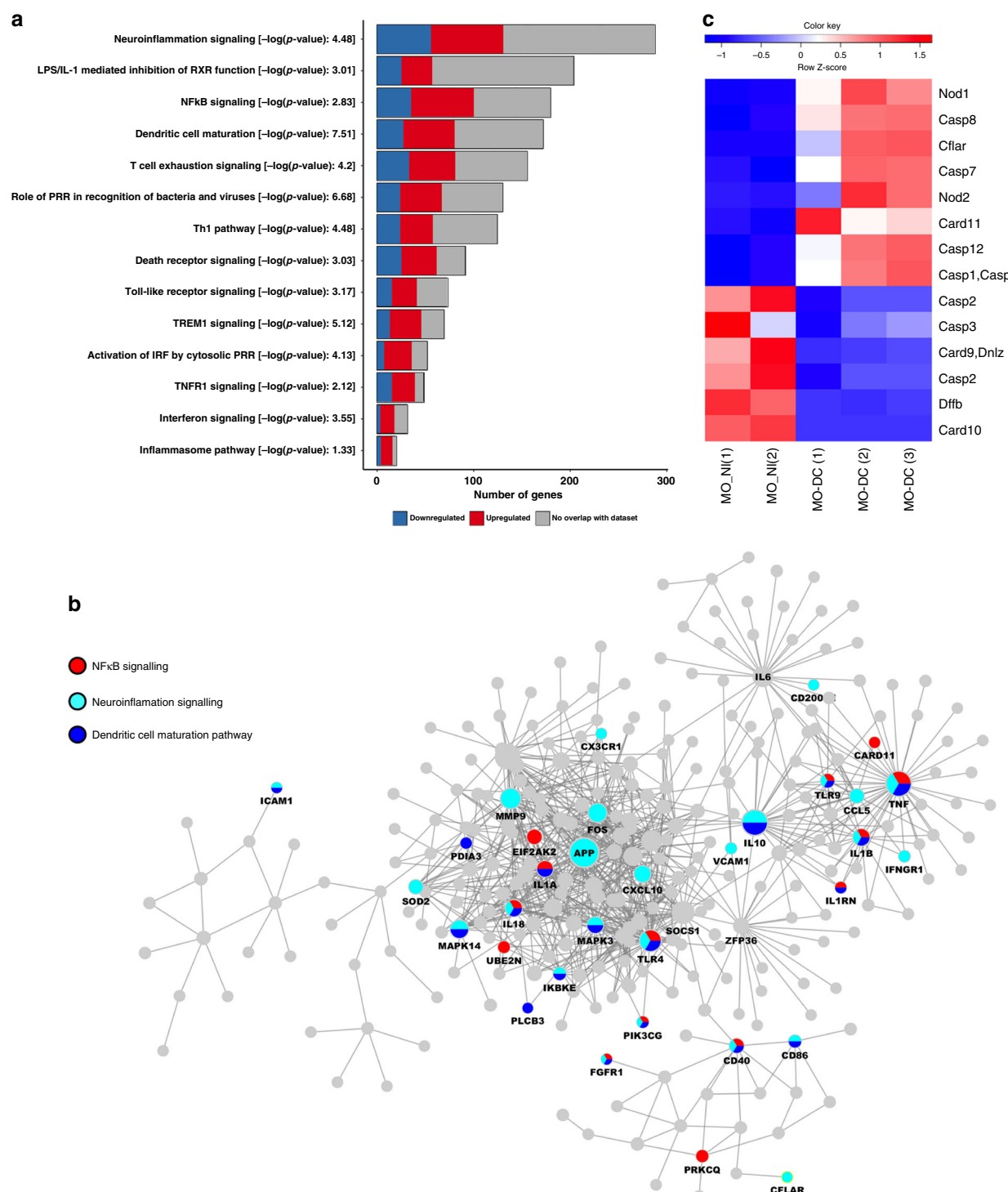

**Fig. 5 Gene signatures from monocyte-derived dendritic cells (MO-DCs) from *PbA*-infected mice.** Monocytes and MO-DCs were sorted from splenocytes from C57BL/6 controls (uninfected) mice and at 5 days post infection for RNA extraction and RNA-seq analysis. **a** Bar plot showing pathways significantly activated in MO-DCs from *PbA*-infected mice. Each color in the bar indicates the number of differentially expressed (DE) genes involved in a given pathway: in blue are downregulated, in red are upregulated, and in gray are the genes that do not show overlap with Ingenuity pathway analysis (IPA) database. **b** The main component of IPA network of all DE genes. All the DE genes are in gray, except the genes from NF-κB signaling, neuroinflammation, and dendritic cell differentiation pathways, which have been highlighted of red, light blue, and dark blue, respectively. **c** Heatmap illustrates the expression of genes from caspase family in highly purified (over 98% purity) splenic MO-DCs (CD11b⁺F4/80⁺CD11c⁺DC-Sign⁺MHC II⁺) from *PbA*-infected C57BL/ 6 mice compared to splenic monocytes (MO, CD11b⁺F4/80⁺CD11c⁻DC-Sign⁻MHC II⁻) from uninfected controls (NI). The color scheme represents the row *Z* score.

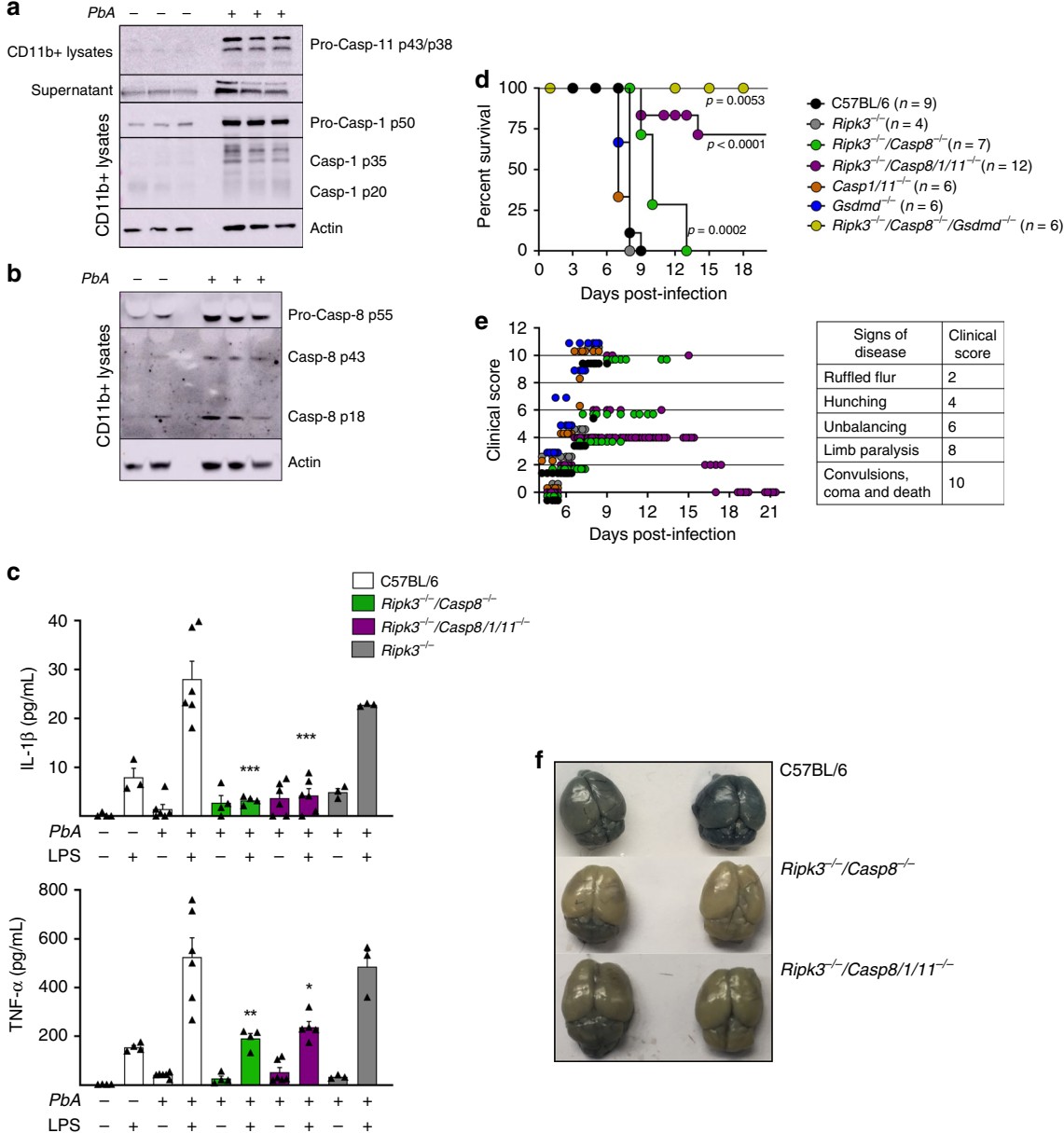

**Fig. 6 Caspase-8 mediates experimental cerebral malaria (ECM) in *PbA*-infected mice.** C57BL/6, *Casp1/11*⁻/⁻, *Gsdmd*⁻/⁻, *Ripk3*⁻/⁻ (control), *Ripk3*⁻/⁻/*Casp8*⁻/⁻, *Ripk3*⁻/⁻/*Casp8/1/11*⁻/⁻, and *Ripk3*⁻/⁻/*Casp8*⁻/⁻/*Gsdmd*⁻/⁻ mice were used in different experiments as indicated. CD11b⁺ cells were purified from splenocytes harvested on days 0 and 5 post infection. Cells were lysed in RIPA buffer to prepare the lysates, and splenocytes were spin down 400 × *g* for 5 min to prepare supernatants. Western blots were revealed with (**a**) anti-caspase-1 and anti-caspase-11 or (**b**) anti-caspase-8 antibodies. **c** Splenocytes from uninfected controls and *PbA*-infected mice were stimulated in vitro with 1 μg/ml of LPS, and 48 h later the levels of IL-1β (top panel, ***P < 0.0001) and TNFα (bottom panel, *P = 0.0104 and **P = 0.0056) measured in culture supernatants by ELISA. For IL-1β: C57BL/6 (control *n* = 4, +LPS *n* = 3; infected *n* = 6), *Ripk3*⁻/⁻/*Casp8*⁻/⁻ *n* = 4, *Ripk3*⁻/⁻/*Casp8/1/11*⁻/⁻ *n* = 6, and *Ripk3*⁻/⁻ *n* = 3. For TNFα: C57BL/6 (control *n* = 4, infected *n* = 6), *Ripk3*⁻/⁻/*Casp8*⁻/⁻ *n* = 4, *Ripk3*⁻/⁻/*Casp8/1/11*⁻/⁻ (*n* = 6, + LPS *n* = 5), and *Ripk3*⁻/⁻ *n* = 3. **d** Survival of *PbA*-infected mice was followed by 21 days, and in (**e**), each of the classic CM symptoms (ruffled fur, abnormal posture, unbalancing, limb paralysis, convulsion, coma, and death) was given a score (0, 2, 4, 8, or 10). **f** After 6 days of infection, mice were injected i.v. with 0.2 mL of 1% Evans blue (120 mg kg⁻¹ d⁻¹) at 6 days post infection of vehicle-treated mice. One hour later, mice were killed, and brain color was assessed. Naive mice were also injected with Evans blue and used as unstained control. In **a**, **b**, blots are representative of two different experiments. In **c**, mean ± s.e.m of two to three different experiments are shown; statistical analysis by one-way ANOVA. The survival curve was analyzed by log-rank test.

**Caspase-8 mediates *P. berghei* ANKA (*PbA*)-induced ECM.** Different studies have shown that double *Casp1/11*⁻/⁻ mice are still highly susceptible to ECM induced by *PbA* infection[50,51]. In contrast, we found that caspase-8 contributes to the development of ECM. Consistent, with the gene expression analysis, our ex vivo results show that splenic CD11b⁺ cells from *PbA*-infected mice express high levels of pro- and active caspases-1, -11

(Fig. 6a), and -8 (Fig. 6b) proteins. Consistent with the results obtained with *Pc*-infected mice, we found that IL-1β and TNFα produced by splenocytes stimulated with LPS were impaired both in *PbA*-infected *Ripk3*⁻/⁻/*Casp8*⁻/⁻ and *Ripk3*⁻/⁻/*Casp8/1/ 11*⁻/⁻ mice (Fig. 6c). Importantly, we observed a delay lethality of *Ripk3*⁻/⁻/*Casp8*⁻/⁻ infected with *PbA*, whereas *Ripk3*⁻/⁻/*Casp8/ 1/11*⁻/⁻ and *Ripk3*⁻/⁻/*Casp8*⁻/⁻/*Gsdmd*⁻/⁻ mice were highly

resistant to ECM (Fig. 6d). In contrast, *Ripk3−/−, Gsdmd−/−*, or *Casp1/11−/−* mice were as susceptible to ECM as the C57BL/6 mice. The clinical signs of disease correlated with the lethality curves (Fig. 6e). The images presented in Fig. 6f show the Evans blue staining of C57BL/6 mice, but not *Ripk3−/−/Casp8−/−* or *Ripk3−/−/Casp8/1/11−/−* infected with *PbA*. Pro-inflammatory cytokines are known to promote blood–brain barrier permeability, leading to leak and infiltration of mononuclear cells contributing to ECM. Hence, these results are consistent with the hypothesis that the defective cytokine response is contributing to *Casp8* knockout mice resistance to ECM. However, consistent with the results from *Pc*-primed mice, caspase-8 also has an important role in mediating ECM, which seems to be independent of GSDM-D cleavage and IL-1β release. These results indicate that caspase-8 and caspase-1 and -11 also have a complementary role in the pathogenesis of ECM.

**Expression of caspases-4 and -8 in monocytes from malaria patients.** Our previous studies show that infection with either *P. vivax* or *P. falciparum* prime circulating monocytes to express an inflammasome gene signature, form AIM2, NLRP3, and NLRP12 specks, express active caspase-1, and produce high levels of IL-1β, when challenged with LPS[18,19,32]. We now evaluated the expression and activation of the inflammatory caspases-4 and -8 as well as GSDM-D in monocytes from malaria patients. It is well established that cytokinemia and fever coincide with the synchronized burst of infected red blood cells and the consequent release of parasites[1]. Hence, blood samples from febrile malaria patients were collected for peripheral blood mononuclear cells (PBMC) isolation. Western blots of monocytes purified of PBMCs from *P. vivax*-infected patients showed active caspase-4 (p32 subunit) (Fig. 7a). In addition, infection with *P. vivax* induced expression and cleavage of caspase-8 in circulating CD14+ monocytes (Fig. 7b). Furthermore, we found that GSDM-D is cleaved in monocytes from *P. vivax*-infected individuals, and not in monocytes from HD (Fig. 7c). A similar pattern of caspase--8, GSDM-D and caspase-4 activation was observed in monocytes from *P. falciparum*-infected patients (Fig. 7d, e). Altogether, we provide evidence that *P. vivax* and *P. falciparum* infection leads to induction and activation of caspases-4 and -8 as well as GSDM-D in monocytes, which may contribute to the overall amount of IL-1β and pathogenesis of malaria.

## Discussion

*Plasmodium* infection results in activation of NLRs, inflammasome oligomerization, and caspase-1 via the canonical pathway in circulating and splenic monocytes. Upon secondary stimuli, these armed cells produce extremely high levels of IL-1β, which is thought to be a key cytokine-mediating high peak of fever as well as high sensitivity to endotoxic shock induced in bacterial superinfection[4–6,18,19]. However, the role of caspase-11 and the noncanonical inflammasome activation[21] has not been taken into consideration. Furthermore, different studies demonstrate the role of caspase-8 on IL-1β synthesis and host cell death in bacterial and fungal infections[27–30]. Here, we report that caspase-11 and the homologous inflammatory human caspase-4, as well as caspase-8, are highly activated during *Plasmodium* infection, both in mouse and malaria patients. We found that while caspase-11 has a limited role, caspase-8 is essential for IL-1β release and to mediate hypersensitivity to septic shock and cerebral disease in ECM model.

Catalyzing the formation of β-hematin crystals (hemozoin) is a key strategy to detoxify the heme released from hemoglobin digestion by *Plasmodium* parasites[52]. Different studies have reported that phagocytosis of hemozoin results in caspase-1

activation and IL-1β secretion by monocytes/macrophages, and that this process is dependent on AIM2, NLRP3, and ASC oligomerization[4–6]. Indeed, it was found that DNA bound to hemozoin is essential to this process by activating TLR9 and inducing expression of inflammasome components as well as pro-IL-1β, whereas hemozoin crystals translocate to the cytoplasm of phagocytic cells and promote inflammasome assembling[6]. In addition, studies performed in our laboratory[19] and elsewhere[53] have shown that immunocomplexes-containing DNA present in plasma of malaria patients as well as opsonized infected RBCs also trigger the formation of inflammasome specks in human monocytes. However, most of the early studies that evaluated the role inflammasome in rodent malaria used a mouse strain that is double deficient for caspase-1 and caspase-11. Importantly, it was found in other infectious disease models that caspase-11 is involved in induction of pyroptosis, alarmin as well as IL-1β secretion, in a process that is independent of the canonical activation of inflammasome[21,54,55]. Hence, we revisited this question to evaluate the contribution of the noncanonical pathway on IL-1β release during malaria.

The inflammasome activation through the noncanonical pathway requires as starting point caspase-11 activation in the mouse. Once activated, these inflammatory caspases are thought to promote potassium efflux, NLRP3, and ASC oligomerization and consequently, caspase-1 activation that amplifies cleavage of pro-IL-1β[56–58]. Different studies also demonstrate that pro-caspase-11 is induced by Type I IFN as well as TLR agonists, such as LPS, Poly IC, or PAMCys3[34–37]. Recent studies have demonstrated that, like caspase-11, cytosolic LPS is an agonist for the noncanonical inflammasome in humans and revealed that the inflammatory caspase-4 is a direct sensor of bacterial endotoxin[22,57,59].

We also found that in vivo *Plasmodium* infection was sufficient to activate caspase-11 in mice. As TLRs have been shown necessary in the process of priming for induction of pro-caspase-11 and TLR7 as well as TLR9 are strongly activated by infection with malaria parasites[31,60,61], we investigated the role of NAS-TLRs on activation of the noncanonical inflammasome pathway. Importantly, activation of NAS-TLRs during malaria is important to trigger IL-12, which is necessary for maximal IFNγ response during acute infection with malaria[31,32,44,62]. We found that expression of pro-caspase-11 was diminished both in triple *Tlr3/7/9* as well as *Ifng* knockout mice infected with *P. chabaudi*. Importantly, our in vitro experiments indicate that whereas IFNγ is required to induce pro-caspase-11, activation with infected erythrocytes promoted cleavage of pro-caspase-11. Indeed, it has been demonstrated that a combined activation by NAS-TLRs and IFNγ results in high expression of pro-caspase-11 in the host cell cytoplasm[41], which initiates the process of self-cleavage[63].

It has also been shown that caspase-11 needs a secondary stimulus with cytosolic LPS to be activated and promote endotoxic shock in a process independent of TLR4[34,35]. It is established that LPS binds to the caspase-11 CARD domain and is required to promote caspase-11 activation. However, our in vivo and ex vivo experiments indicate that caspase-11 was not necessary to cleave caspase-1 and had a limited role in secretion of IL-1β or susceptibility to the low-dose LPS-induced shock. Different from the bacteria models described above, *Plasmodium* infection was sufficient to induce and activate caspase-11, and TLR4 was required to induce pro-IL-1β, the release of IL-1β, and lethality induced by LPS challenge.

In addition to caspase-1 and caspase-11, caspase-8 has also been shown to play a key role in regulating IL-1β secretion in bacterial and fungal infections[27–29]. These studies indicate that caspase-8 mediates the synthesis of pro-IL-1β by activating NF-κB. Unlike caspases-1 and -11, the role of caspase-8 does not

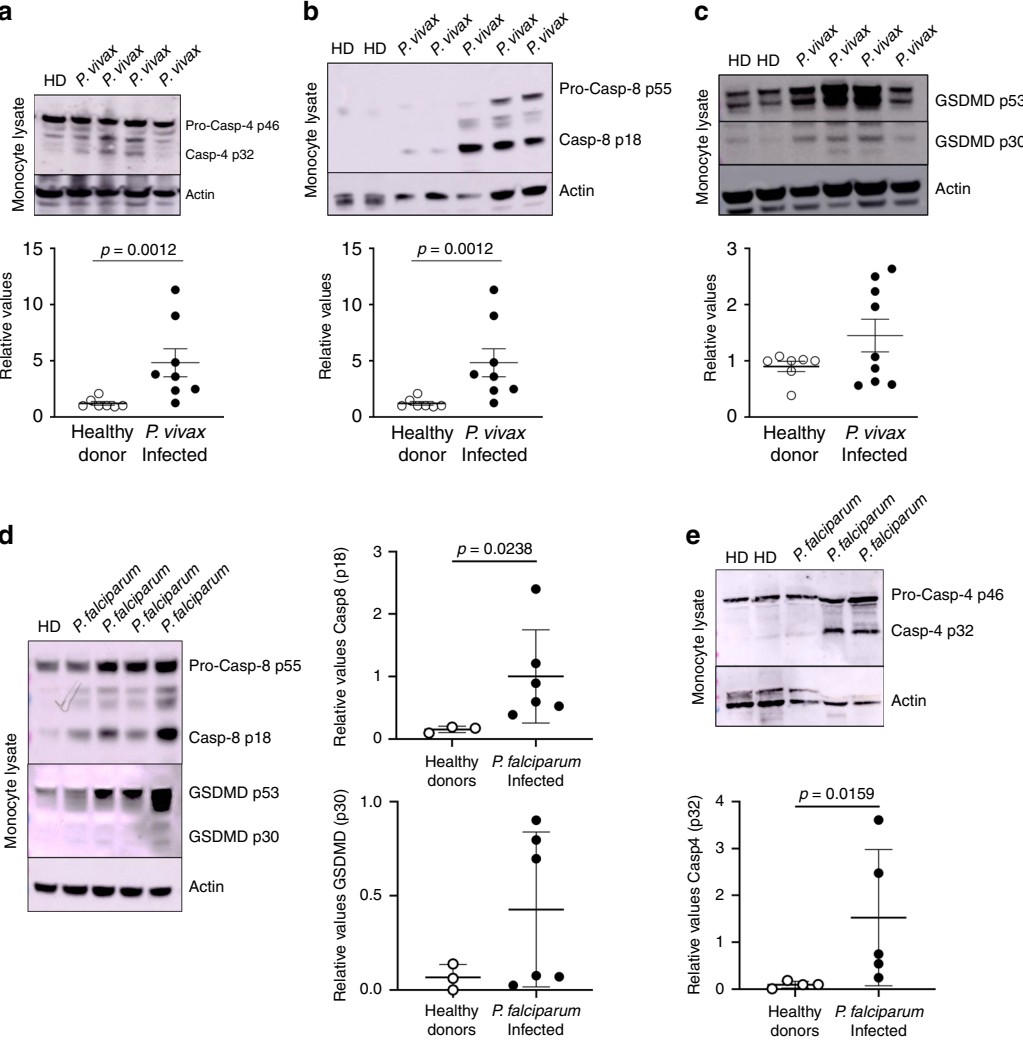

**Fig. 7 Activation of caspases-4 and -8 in monocytes from malaria patients.** CD14+ monocytes were isolated from peripheral blood mononuclear cells (PBMCs) from healthy donors (HD) and malaria patients (*P. vivax*), lysed with RIPA buffer and analyzed by western blot. **a** Western blot was performed with an anti-caspase-4 antibody. The levels of caspase-4 p32 expression were analyzed by densitometry and shown in the bottom panel (HD $n = 7$ and *P. vivax* $n = 8$). **b** Western blot was performed with CD14+ monocytes lysates from two HD and five *P. vivax* malaria patients, and expression of caspase-8 (p55) and cleaved caspase-8 (p18) detected using an anti-caspase-8 antibody. The levels of caspase-8 p18 expression were analyzed by densitometry and shown in the bottom panel (HD $n = 4$ and *P. vivax* $n = 9$). **c** Western blot was performed with CD14+ monocytes lysates from HD as well as *P. vivax* malaria patients and revealed with an anti-GSDM-D antibody. The levels of cleaved GSDM-D p30 expression were analyzed by densitometry and shown in the bottom panel (HD $n = 7$ and *P. vivax* $n = 9$). **d** CD14 + monocytes lysate isolated from healthy donors (HD) and malaria patients (*P. falciparum*) were analyzed by western blot performed against caspase-8 and GSDM-D. The levels of caspase-8 and GSDM-D expression were analyzed by densitometry and shown in the side panels (HD $n = 3$ and *P. falciparum* $n = 6$). **e** CD14 + monocytes lysate isolated from healthy donors (HD) and malaria patients (*P. falciparum*) were analyzed by western blot performed against caspase-4. The levels of caspase-4 expression were analyzed by densitometry and shown in the bottom panel (HD $n = 4$ and *P. falciparum* $n = 5$). Statistical analysis was performed by two-tailed unpaired Mann–Whitney test at 95% CI. Densitometry graphs show mean ± s.e.m of samples from patients run on two to three representative gels with similar results.

seem to be IL-1β specific, since its deficiency also impairs the synthesis of other cytokines, such as TNFα and IL-6[46]. Consistently, we found that the release of both IL-1β and TNFα was highly impaired in the caspase-8-deficient mice infected with *Pc* or *PbA*.

Caspase-8 was also shown to be involved in GSDM-D cleavage[30]. However, in our experiments, caspase-8 was not required for caspase-1 or caspase-11 cleavage. Importantly, the *Pc*-infected *Gsdmd*[−/−] phenocopied the *Casp1*[−/−] mice, which were slightly more resistant to the LPS-induced shock. Intriguingly, *Plasmodium*-infected *Gsdmd*[−/−] mice showed severe impairment of IL-1β release, but remained susceptible to the low-dose LPS challenge or ECM. Therefore, the lethality in these models cannot be

explained solely based on cleavage and release of IL-1β. Furthermore, either *Casp1/11*[−/−][50,51] or *Gsdmd*[−/−] mice were highly susceptible to ECM. Nevertheless, like *Ripk3*[−/−]/*Casp8/1/11*[−/−], the *Ripk3*[−/−]/*Casp8*[−/−]/*Gsdmd*[−/−] mice were highly resistant to ECM. Together, these results indicate that while redundant in these malaria models, the role of caspase-8 is complementary but distinct from the canonical inflammasome that involves caspases-1 and GSDM-D activation.

Infection with *P. vivax* elicits high fever and paroxysm thought to be dependent on IL-1β, whereas *P. falciparum* infection is the most lethal species and the cause of cerebral malaria in humans, which is dependent on TNFα[47]. Hence, we evaluated whether inflammatory caspases and GSDM-D are activated in monocytes

from malaria patients. We have previously shown that AIM2 and NLRP3 inflammasomes, as well as caspase-1, are activated in monocytes from malaria patients[18]. Here, we found that in vivo infection with either *P. vivax* or *P. falciparum* is sufficient to activate caspase-4 and cleave GSDM-D in monocytes without exposure to LPS. In addition, we found that monocytes from patients infected with *Plasmodium spp.* process caspase-8, indicating their potential role in IL-1β release and pathogenesis of malaria.

In conclusion, our results demonstrate that caspase-1, caspase-8, the mouse caspase-11, and its human ortholog caspase-4 are all activated in monocytes from *P. vivax* and *P. falciparum* malaria patients and mouse malaria models. The important role of the canonical activation of the inflammasome is revealed in caspase-8-deficient mice. While redundant, our data indicate that in these malaria models the mechanism by which caspases-1/11 promotes IL-1β release involves GSDM-D, whereas caspase-8 seems to mediate expression of pro-IL-1β and TNFα. As caspase-8 controls the release/expression of two key cytokines on *Plasmodium*-induced disease, it should be considered a potential target for therapeutic intervention in malaria patients.

## Methods

**Ethics statement**. The Ethical Committee on Human Experimentation from Centro de Pesquisas em Medicina Tropical (CEP-CEPEM 096/2009), the Brazilian National Ethical Committee (CONEP 15653) from Ministry of Health and the Institutional Review Board from the University of Massachusetts Medical School (UMMS, IRB-ID11116) approved this study performed with malaria patients. Experiments with mice were conducted according to institutional guidelines for animal ethics and approved by the Institutional Ethics Committees from Oswaldo Cruz Foundation (Fiocruz-Minas, CEUA/LW15/14, and LW16/18) and UMMS (IACUC/A-2371-15), respectively.

**Malaria patients**. Healthy donors volunteers ($n = 8$) from Porto Velho or Belo Horizonte were used as negative controls. Symptomatic febrile patients with acute uncomplicated *P. vivax* ($n = 9$) and *P. falciparum* malaria ($n = 6$) were recruited at CEPEM, an outpatient malaria clinic in Porto Velho, an endemic malaria area in the Amazon region of Brazil. Written informed consent was obtained before enrollment of all subjects. Up to 100 mL of peripheral blood was collected immediately after confirmation of *P. vivax* or *P. falciparum* infection by a standard thick blood smear. Peripheral blood was also collected from healthy donors living in an endemic area.

**Experimental infections**. All mouse lineages used in this study have been back-crossed for at least ten generations into the C57BL/6 genetic background. *Casp11*−/−, *Casp1*−/−/11tg, and *Gsdmd*−/− were provided by Dr. Vishva Dixit from Genentech (San Francisco, CA). *Caspase-1*−/−/11−/− and *Tlr3*−/− mice were provided by Dr. Richard Flavell from Yale University (New Haven, CT). The *Casp1*−/− mice were provided by Dr. Devi Kanneganti Thirumala from St. Jude Children's Research Hospital (Memphis, TN). *Ripk3*−/−, *Ripk3*−/−/*Casp8*−/−, and *Ripk3*−/−/*Casp8/1/11*−/− were provided by Egil Lien. The *Tlr9*−/−, *Tlr4*−/−, and *Tlr7*−/− mice were provided by Dr. Shizuo Akira from Osaka University (Osaka, Japan). The *Ripk3*−/−/*Casp8*−/−/*Gsdmd*−/− and *Tlr3/7/9*−/− mice were generated in our laboratory by genetic crosses. The C57BL/6, *Ifng*−/−, *Tnfr*−/−, *Ifnar1*−/−, and 129S6 were originally obtained from Jackson Labs. All mouse lineages mentioned above were bred and maintained in microisolators at Fiocruz-Minas and UMMS on a 12 h dark/light cycle, temperature range was 68–79 °F, and humidity between 30 and 70%. Female and male mice between 6 and 10 weeks old were used in all experiments.

The *P. chabaudi chabaudi AS* and *P. berghei* ANKA strains were used for experimental infection and kept in our laboratory as previously described[31,64]. Briefly, *P. chabaudi* and *P. berghei* strains were maintained in C57BL/6 mice by passages once and twice a week, respectively. For experimental infection, wild-type and various knockout mice were infected i.p. with $10^5$ infected red blood cells diluted in PBS 1×. In some experiments, mice 0 (uninfected controls) and 8 days post infection with *P. chabaudi* were challenged with LPS (serotypes O55:B55 or 0111:B4 from *E. coli* —Sigma and InvivoGen, respectively) and plasma collected 8 h later. LPS was used at a concentration 10 μg/mouse, as indicated. In experiments with *P. berghei* ANKA, mice were followed daily for survival and to score clinical symptoms. ECM signs included ruffled fur, abnormal postural responses, reduced reflexes, reduced grip strength, coma, and convulsions. Mice that demonstrated complete disability in all parameters or died between days 7 and 12 post infection were considered as having ECM[31,44].

**Human monocyte lysates**. PBMCs were isolated from whole blood on Ficoll-paque Plus (GE Healthcare). Monocytes were purified from PBMCs of *P. vivax* and *P. falciparum*-infected patients or healthy donors by using immunomagnetic beads for positive selection of CD14+ cells (Miltenyi Biotec). Cells were lysed with RIPA buffer (Sigma) solution with a protease inhibitor cocktail (ThermoFisher).

**Splenocytes and CD11b+ cells**. Spleens from control and infected mice were passed through a 100-μm nylon cell strainer. After the first centrifugation (5 min $400 \times g$), tissue supernatant was collected. Erythrocytes were then lysed, washed, and splenocytes resuspended in RPMI 1640 medium supplemented with penicillin, streptomycin, and 10% fetal bovine serum (FBS) (Gibco, ThermoFisher). In some experiments, splenocytes ($2 \times 10^6$ cells) were stimulated in vitro with 1 μg/ml of LPS for 24–48 h. For experiments with CD11b+ cells purified from splenocytes, immunomagnetic beads for positive selection of CD11b+ cells (Miltenyi Biotec) were used.

**Bone marrow-derived monocytes (BMDMs)**. Bone marrow cells from C57BL/6 mice were cultured in DMEM (Corning) with 10% FBS, 10 mM L-glutamine (Gibco, ThermoFisher), and 20% L929 supernatants. After 7 days $2 \times 10^6$ BMDMs were cultured in 48-wells plate in 300 μl of medium without FBS. The BMDM were stimulated with 1:5 of red blood cells (RBC) or infected RBCs (iRBC), 40 ng/ml of IFNγ (Sigma), and 500 U/ml of IFN-I (Pbl Assay Science), as indicated in the figure legends. Between 16 and 24 h of in vitro stimulation, BMDMs were lysed with RIPA buffer (Sigma) solution. Culture supernatants were collected, and the soluble proteins were precipitated using chloroform (25% of supernatant volumes) and methanol (same supernatant volumes). After centrifugation at $20,000 \times g$ for 10 min at 4 °C, the proteins pellet was dried in 56 °C and suspended in sample buffer.

**Western blotting**. Total of mouse splenocytes, CD11b+ cells, BMDM, or monocytes from patients were lysed with RIPA buffer solution with a protease inhibitor cocktail. After 15 min on ice, lysates were centrifuged at $13,000 \times g$ for 10 min at 4 °C. The proteins were separated in a 12%-acrylamide (caspase-11 and caspase-8) or 4–12% acrylamide (caspase-1/caspase-4/GSDM-D) NUPAGE Bis-tris Protein gels (Invitrogen) and transferred onto nitrocellulose membranes. The membranes were incubated with caspase-11 (Novus Biologicals, 1:1000), caspase-1 (Adipogen, 1:1000), caspase-4 (Cell Signaling, 1:1000), anti-mouse and anti-human caspase-8 (Enzo Life Science, 1:1000), anti-mouse cleaved caspase-8 (Cell Signaling, 1:500), anti-GSDM-D (sigma, 1:1000) and β-actin (Sigma, 1:2000) specific antibodies, then incubated with HRP-conjugated secondary antibodies (Jackson ImmunoResearch, 1:25,000) and detected with Clarity Max ECL Substrate (Biorad) using ImageLab Touch Software V6.0.1 (Bio-Rad). Bands quantification was performed with ImageStudio V5.2. Uncropped western blot is available in the Supplementary Information file (Supplementary Figs. 5–11).

**Cytokine measurements**. Measurements of mouse IL-1β and TNFα in plasma or supernatant of splenocyte cultures were performed using commercially available ELISA Duoset kit (ThermoFisher). Absorbance was read with SOFTmaxPRO V4.3.1 LS.

**Flow cytometry**. Splenocytes ($5 \times 10^6$ cells) from mice at 0 (uninfected controls) and 8 days post infection were harvested 2 h after challenge with LPS and stained with fluorescein-labeled monoclonal antibodies (mAbs) specific for cell surface markers or cytokines. The following flow cytometry mAbs specific were used: CD11c-Alexa fluor 700 (1:100), MHC II-APCCy7 (1:200), F4/80- APC (Biolegend, 1:200), CD11b-PECy7 (1:4000), pro-IL-1β-FITC (eBioscience, 1:100), and AmCyan (Live/Dead kit, ThermoFisher). The intracellular fixation and permeabilization buffer set of eBioscience (ThermoFisher) were used to perform pro-IL-1β stain.

**Cell sorting**. Spleens from the C57BL/6 control uninfected and *PbA*-infected mice, at 5 days post infection, were harvested, and splenocytes stained with Ly6G (FITC, 1:400) CD11b (PECy7, 1:4000), F4/80 (PECy5, 1:200), CD11c (Alexa fluor 700, 1:100), MHC II (APCCy7, 1:200), and DC-SIGN (APC e-fluor 660, 1:800) and then submitted to purification by using a cell sorting ARIA (BD). These cells were first gated on FSC-H/FSC-A, to avoid doublets. Next, we gated on Ly6G−, then CD11b+F4/80+. Cells from noninfected controls were then gated on MHC II- DC-SIGN- and from infected mice on MHC IIhigh DC-SIGNhigh. The gated cells were sorted, collected into fresh new tubes, the MO-DCs were confirmed to be CD11c+ and then used for RNA extraction and RNA-Seq.

**RNA-Seq samples and library preparation**. RNA-seq was performed in biological replicates (three mice per group). Monocytes and MO-DCs were purified from splenocytes using a cell sorting ARIA (BD). Since *PbA* infection leads to severe cerebral symptoms and death after the 6th day of infection, MO-DCs (CD11b+F4/80+CD11c+DC-Sign+MHC II+) were collected from *PbA*-infected C57BL/6 mice at day 5 post infection and monocytes (CD11b+F4/80+CD11c−DC-Sign−MHC II−) collected from noninfected mice. The total RNA was extracted using the Qiagen RNeasy Mini Kit (Qiagen), and was cleaned and treated with DNase I (Qiagen). RNA-seq libraries were prepared using the TruSeq Stranded mRNA Kit (ILLUMINA)

following the manufacturer's instructions. The total RNA and library quality were verified by fragmentation analysis (Agilent Technologies 2100 Bioanalyzer) and submitted for sequencing on the Illumina NextSeq 500 (Bauer Core Facility Harvard University).

**RNA-Seq analysis.** Reads previously trimmed with Trimmomatic (Bolger et al.[65]) were mapped to the Genome Reference Consortium Mouse Build 38 patch release 5 (GRCm38.p5) using STAR aligner (Dobin et al.[66]) and the Fragments Per Kilobase of transcript per million mapped read (FPKM) values and the differential analysis were calculated with CUFFLINKS (Trapnell et al.[67]). Genes which were differentially expressed in response to the infection with fold change ≥ 1.5 were analyzed for pathway enrichment and network analysis with IPA– Ingenuity Pathway Analysis from Qiagen (USA); and graphs were done in R statistical environment, version 3.4.3 (R Core Team, 2019) and Cytoscape software (Shannon et al.[68]).

**Reagents and softwares.** Please see Supplementary Table 1.

**Statistical analysis.** All data were analyzed using Graphpad Prism V7.0 Software. The differences between two groups were verified using $t$ test or the Mann–Whitney test for parametric or nonparametric data, respectively, at 95% confidence interval (CI). For analysis of multiple groups, we used one-way or two-way analysis of variance (ANOVA) with additional Bonferroni's test at 95% CI. Differences were considered statistically significant when the $P < 0.05$.

**Reporting summary.** Further information on research design is available in the Nature Research Reporting Summary linked to this article.

## Data availability

All data generated or analyzed during this study are included in this published article (and its supplementary information files). RNA-seq data are available at the Gene Expression Omnibus Database, GEO accession number is: GSE126381. All data from this study are available in figshare with the identifier: https://doi.org/10.6084/m9. figshare.12751163.v1. Source data are provided with this paper.

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

## Acknowledgements

We thank Dr. Milton Pereira for his assistance on this paper. We are grateful to the Program for Technological Development in Tools for Health–PDTIS its facilities at Fiocruz-Minas, to the clinic, laboratory and administrative staff, field workers, and subjects who participated in the study. This work was supported by the National Institutes of Health (R01NS098747, R01AI079293, and U19 AI089681—Amazonian Center of Excellence in Malaria Research); Rede Mineira de Biomoléculas from Fundação de Amparo à Pesquisa de Minas Gerais (Fapemig, RED-00012-14), Brazilian National Institute of Science and Technology for Vaccines granted by Conselho Nacional de Desenvolvimento Científico e Tecnológico (CNPq)/Fundação de Amparo à Pesquisa do Estado de Minas Gerais (Fapemig)/Coordenação de Aperfeiçoamento de Pessoal de Ensino Superior (CAPES) (465293/2014-0). N.M.A. received CAPES and LMNP CNPq/ CAPES fellowships. R.T.G. and D.S.Z. are research fellows from CNPq.

## Author contributions

Wrote the paper: R.T.G., L.M.N.P., and D.T.G. Designed experiments and analyzed the results: R.T.G., L.M.N.P., P.A.A., N.M.A., D.S.Z., and D.F.D. Performed experiments: L. M.N.P., P.A.A., N.M.A., D.F.D., C.J., and M.A.A. Contributed with reagents/materials/ samples: D.B.P., K.A.F., E.L., and D.S.Z.

## Competing interests

The authors declare no competing interests.
