## [Peer Review File · Nature Communications]

Reviewers' Comments:

Reviewer #1:

Remarks to the Author:

The manuscript by Pereira et al., investigates the roles of various caspases, including casp1, 8, 4/11, and other molecules in IL-1b production and inflammation during malaria infections. This study presents more detailed mechanisms compared with their previous reports on inflammasome and related signaling pathways. Casp1 and 8 were shown to be necessary for IL-1b release in two rodent malaria models (*P. chabaudi* and *P. berghei*); Casp11 appears to play a minor role in IL-1b secretion in *P. chabaudi*-infected mice when stimulated with LPS. The whole study was designed based on known inflammasome activation pathways worked out in other systems previously. Therefore, this study provides inflammasome activation and IL-1b production specific for malaria infections. The majority of experiments were derived from genetic knockout mice and rodent parasite infections, although data from patients infected with the human malaria parasite *P. vivax* were also presented. The results also show differences in the malaria models; *P. chabaudi* infection required LPS stimulation, but *P. vivax* and *P. berghei* alone can activate related caspases. It is actually interesting to investigate the differences in inflammasome pathways after infection of different malaria parasite species. Although the manuscript is generally well organized, there are places of obvious typos and careless mistakes.

Most of the experiments were done from mice 8 days post infection, whereas the RNA-seq was done from cells 5 days post infection. How and why these days were determined? Inflammasome signaling is generally activated quickly. In human *P. vivax* infection, the exact times of infections are not known.

Minor specific points:

- 1, Page 5, second para, line 2: The sentence is not complete.
- 2, Page 6, second para, line 7, IFNg, but not IFN type I: There are pro-casp11 bands in the lysate after IFN-I stimulation.
- 3, Page 6, second para, line 10, from single TLR3, TLR7, TLR9 deficient mice?
- 4, Page 8, line 3, IFN-AR^B-/-.
- 5, Page 9, line 1-2: Figure 3C and 3D switched.
- 6, Page 9, third para, line 2: expression pro-caspase-8: pro-caspase-8 expression.
- 7, Page 10, line 3-4, "As observed for IL-1 β , TNF α production by splenocytes from Pc-infected mice was impaired": Mice with various gene deficiency were impaired?
- 8, Page 10, line 7: Insert Figure 4k after "Pc-infected TNFR^{-/-} mice".
- 9, Page 10, second para, line 5 Figure 4A (B,C): should be Figure 5a (b,c).
- 10, Page 11, Figure 4A-4E should be Figure 6a-6e (also use 'a' instead 'A'). The manuscript appears to be originally formatted for another journal.
- 11, Page 16, second para, line 6, "in Pc-infected mice that are hyper-susceptible to septic shock and ECM models": Insert " in *P. berghei*" before ECM because Pc generally does not induce ECM.
- 12, Page 20, second para, line 7: what is EL?
- 13, Page 21, second line from the bottom: 20.000.
- 14, Page 22, line 2, ripa, Ripa and RIPA were used.
- 15, Page 24: GSE126381, not able to find the data.
- 16, References: missing info for 30, 42, and 49.
- 17, Figure 1: A, lane labeling was shifted; C, why the cleaved p43 and p18 have difference intensities? Two-tailed nonparametric unpaired t-test, not sure there is a non-parametric t-test? Mann-Whitney?
- 18, Figure 2A: The bottom bands should be b-actin?
- 19, Figure 3B, left: missing labels for the bars.
- 20, Figure 4C: why higher MFI-Pro-IL-1b in the casp1/11^{-/-} group? 4E top, why Casp8 p55 in non-

infected? 4E bottom, no casp8 p43 in one of the casp1/11-/- infected mouse or mis-labeling?
21, Figure 4, nonparametric one-way ANOVA: Kruskal-Wallis?
22, There are many left/right and top/bottom panels in the figures. It is better to simply label them with a, b, c, sequentially.

Reviewer #2:

Remarks to the Author:

In this paper the authors describe a novel role for dendritic cell derived caspase 8 in the pathogenesis of cerebral malaria. They nicely show that caspase 8 processes IL1b and this contributes to the pathology of the disease. More perplexing, however, their data suggests much of this phenotype is independent of gasdermin D, but involves transcriptional regulation of TNFa to synergize with IL-1b to induce deleterious effects. These data are interesting and the authors hypotheses are supported by an extensive set of experiments. There are some questions that arise from the data

1. How is caspase 8 driving the IL1b release if it is independent of gasdermin D and RIP3? These observations suggest the working hypothesis of pore formation being required for IL1b release may not be correct, another pore is being formed to release the IL1b or caspase 8 is upregulating another IL1b processing mechanism (this seems unlikely however).

2. In the paper the authors say that "Unlike caspases-1 and -11, the role of caspase-8 does not seem to be IL-1b specific, since its deficiency also impairs the synthesis of other cytokines, such as TNFa and IL-6" Do the authors think that the effect of caspase 8 is mostly related to priming given that caspase 8 can feed into gene regulatory pathways that modify Nfkb?

Overall the paper would benefit from clarity on the mechanisms by which caspase 8 is regulating IL1b production, processing and release.

Minor points

1. "Furthermore, in oppose to IFNg-/- mice 24" should read " Furthermore, opposite to the IFNg-/- mice" p24

2. Fig 2H does not add much as you would expect GasD-/- to not be able to release IL1B

3. Fig 3A lysate needs to be moved on the figure as the text is on the blot

Reviewer #3:

Remarks to the Author:

Overall, MS by Pereira et al is well-written, elegant and brings new insights into malaria immunopathogenesis, particularly with respect to the role of Caspase-8. Nevertheless, there are several points that should be clarified.

General comments:

1- In this study, authors show data on human malaria (*P. vivax*), and two mouse models (*P. chabaudi* and *P. berghei*). Although I do understand the need of mouse models to explore some mechanisms, the purpose of these three sets of experiments and how they are interconnected is not clear. For instance: *P. berghei* normally represents severe malaria, especially cases of cerebral malaria in children caused by *P. falciparum*, *P. chabaudi* has been used to understand immune response to mild malaria and how immunity is raised after multiple infections. Authors include some interesting data on the *P. vivax* caspase mechanism; however, it is difficult to link these data to those obtained from mouse models and understand the meaning of them.

2- The title mentions cerebral malaria, but why show data obtained from *vivax* (which does not

represent severe complications in this study) and *P. chabaudi*?

Specific comments:

1- Abstract: Authors state that this study suggests the mechanism... Does this mean that the mechanism was not elucidated?

2- Introduction: Text should be revised. There are a few errors. Also: "...In another member of the family, caspase 8..." which family is this? It is not clear why is it relevant to investigate the role of these caspases in malaria pathogenesis? What is the role of caspase 5?

3- Results: I believe patient data should go in the end, to confirm the relevance of data obtained from mouse model. They state that caspases 4, 5 and 8 have been activated in vivax infection. Is this correct? If so, this is not clear. In fact, it seems that the vivax data only shows that caspases are activated, but it is not specific. Maybe the authors could better explore *P. vivax* monocyte activation by stimulating ex vivo with parasite products instead of LPS. "The results presented in Figures 4A show that the levels of circulating IL-1 β on RIP3^{-/-}-caspase-8^{-/-} or RIP3^{-/-}-caspase-8/1/11^{-/-} mice" Why not the single Caspase 8 KO?

4- Discussion: it is stated: "We found that while caspase-11 has a limited role, caspase-8 is essential for IL-1 β release, mediates hypersensitivity to septic shock and cerebral disease in experimental malaria model" However, in ECM IL-1 β , this does not seem relevant.

RESPONSE TO REFEREES NCOMMS-19-05667 – May 7, 2020

Reply to the Editor and three Reviewers' main comments:

*Reviewer 1 has done a fair description of our work, highlighting its importance in the malaria context. Malaria, together with AIDS and Tuberculosis, is one of the most devastating infectious diseases in the world. A hallmark of malaria is the paroxysm, which is largely attributed to the pyrogenic cytokine IL-1 β . It was a surprise that previous studies demonstrated that genetically deficient mice that do not form inflammasome and release IL-1 β had no obvious disease phenotype in the experimental cerebral malaria model (ECM). This was only observed when mice were co-infected with Gram-negative bacteria, which are important co-factors for severe malaria (Ataide et al., 2014). Indeed, in mice acutely infected with *Plasmodium sp.* there is a well-documented pro-inflammatory priming that is central to the pathogenesis of malaria. Due to this pro-inflammatory priming malaria patients become highly susceptible to bacterial infection. This can be reproduced in the malaria mouse models that become very susceptible to a very low dose (10 μ g) of LPS challenge. At least, 500 μ g of LPS is required to induce lethality in non-infected mice. To address this question in a more holistic way, we investigated, simultaneously, the role of the canonical, non-canonical (caspase-11) and caspase-8 pathways in two different malaria models. As a note, there are important crosstalk of these three pathways involved on IL-1 β release.*

The previous studies performed in our lab (Ataide et al., 2014) and elsewhere addressed the role of the canonical pathway in the malaria context. However, after the discovery of caspase-11 as a non-canonical activator of inflammasomes, the whole literature of inflammasome activation had to be revised. It was found that in certain bacterial infection caspase-11 has an important role in IL-1 β release and pathogenesis. To our knowledge the role of caspase-11 on IL-1 β release has not been addressed in malaria and many other parasitic infections. Thus, we now demonstrate that while both caspase-1 (as previously shown) and caspase-11 are activated during malaria, the former caspase, rather than caspase-11, has the dominant role on IL-1 β secretion.

Still, no disease phenotype is observed in the ECM model in mice deficient on both caspase-1 and caspase-11. A phenotype is only observed, if we used the low dose-LPS challenge to mimic the superinfection with Gram-negative bacteria. As pointed out by Reviewers 2 and 3, the second and more important part of our study is the finding that caspase-8 has an important role on malaria pathogenesis. Importantly, in deficiency of caspase-8, we can now appreciate an important role for caspase-1 in the ECM or in the septic shock model. As previously described, the deficiency of caspase-8 results in a partial impairment of IL-1 β release. Thus, one could claim that an additive effect of caspase-1 and caspase-8 deficiencies on impairment IL-1 β release is responsible for the enhanced resistance to ECM and LPS challenge. In addition, to be required for optimal IL-1 β release, caspase-8 also regulates the release of other cytokines, including TNF α that is known to play a central role on malaria pathogenesis. Hence, as indicated by Reviewer 2, we favor the hypothesis that the mechanism of enhanced resistance to disease in mice deficient in both caspase-1 and caspase-8 is due to the combined impairment of IL-1 β and TNF α release.

New Data

Recent studies suggest that caspase-8 crosstalk with the inflammasome pathway and mediate pyroptosis by cleaving GSDM-D. To further establish that caspase-8 and caspase-1/11 were acting by different mechanisms, we generated the RIP3/Caspase-8/GSDM-D triple knockout mice. Our results show that in contrast to caspase-1/11^{-/-}, GSDM-D^{-/-} or the partially resistant Caspase-8^{-/-}, the RIP3/Caspase-8/GSDM-D^{-/-} mice are highly resistant to ECM. While somewhat redundant, our results also suggest that caspase-8 and caspases-1/11 (and GSDM-D) are acting in distinct pathways that seem to be complementary to each other in mediating ECM (Figure 4). Our data indicate that while caspase-8 is controlling the expression and release of IL-1 β and TNF α , caspase-1/11 and GSDM-D are mediating the pathogenesis of malaria by regulating pyroptosis and IL-1 β release.

Regarding the data of malaria patients, we now include the requested immunoblots for the various inflammatory caspases and GSDM-D with samples of patients infected with *P. falciparum*, the etiological agent that causes cerebral malaria (Figure 5).

Specific Comments:

Reviewer 1:

“Most of the experiments were done from mice 8 days post infection, whereas the RNA-seq was done from cells 5 days post infection. How and why these days were determined? Inflammasome signaling is generally activated quickly. In human *P. vivax* infection, the exact times of infections are not known.”

1. These are two different models and the timing is different. In the *P. chabaudi* model the peak of the pro-inflammatory response (cytokinemia) is at day 7-8 post-infection. In the cerebral malaria model the peak of the inflammatory response is in day 5-6. The mice become very sick and succumb to infection on days 6 and 7 (see **Figure 4D**). Hence, we normally analyze the inflammatory response in the latter model at day 5 post-infection. This is now indicated in the **legend of Figure 3** (RNA-Seq) and **Figure 4** (immunoblots and cytokines) of the cerebral malaria model as well as on the **RNA-Seq Method Details** (RNA-Seq samples and Library Preparation).
2. It is hard to determine the time of infection in the *P. vivax* malaria patients since this is not an experimental infection. There is an obvious incubation period, which is thought to vary from 7 to 15 days depending on parasite isolates and host factors. In our studies, we collect samples from febrile patients that occur soon after red blood cells burst, releasing parasites at the parasitemia peak. It is well established that cytokinemia coincides with the synchronized burst of infected red blood cells. This is now clarified in the results section of **Figure 5** and on the **Malaria Patients section**.
3. We apologize for all the typos and thank the **Reviewer** for carefully revising the text and Figures of our manuscript. They are now corrected in the new version of the manuscript.

Minor specific points:

Point 1 - Page 5, second para, line 2: The sentence is not complete.

The sentence is now completed. “To study the importance of this pro-inflammatory priming on malaria pathogenesis, we used a low-dose LPS challenge” (page 5 line 4-5)

Point 2 - Page 6, second para, line 7, IFN γ , but not IFN type I: There are pro-casp11 bands in the lysate after IFN-I stimulation.

The sentence is now corrected to "IFN γ and IFN Type I are sufficient to induce" (page 6 line 12).

Point 3 - Page 6, second para, line 10, from single TLR3, TLR7, TLR9 deficient mice?

The sentence is now corrected to "TLR3, TLR7 and TLR9 deficient mice" (page 6 lines 15-16).

Point 4 - Page 8, line 3, IFN-AR β -/-

It is now corrected to "IFN-AR β -/-" (page 7 line 21).

Point 5 - Page 9, line 1-2: Figure 3C and 3D switched.

The order of the images is now corrected on **Supplementary Figures 1C and 1D**.

Point 6 - Page 9, third para, line 2: expression pro-caspase-8: pro-caspase-8 expression.

The sentence is now corrected to "pro-caspase-8 expression" (page 8 line 23).

Point 7 – Page 10, line 3-4, “As observed for IL-1 β , TNF α production by splenocytes from Pc-infected mice was impaired”: Mice with various gene deficiency were impaired?

The sentence is now corrected to "As observed for IL-1 β , TNF α production by splenocytes from Pc-infected mice was impaired in mice with various gene deficiencies" (page 9 lines 8-9).

Point 8 – Page 10, line 7: Insert Figure 4k after “Pc-infected TNFR β -/- mice”.

The figure number (**Figure 2K**) is now inserted in the sentence (page 9 line 13).

Point 9 – Page 10, second para, line 5 Figure 4A (B,C): should be Figure 5a (b,c).

The number of all the figures is now corrected. The letters are now changed from capital to lower case.

Point 10 - Page 11, Figure 4A-4E should be Figure 6a-6e (also use ‘a’ instead ‘A’).

The number of all the figures is now corrected. The letters are now changed from capital to lower case.

Point 11 - Page 16, second para, line 6, “in Pc-infected mice that are hyper-susceptible to septic shock and ECM models”: Insert “ in P. berghei” before ECM because Pc generally does not induce ECM.

After re-writing the Discussion topic this sentence is no longer present on the manuscript.

Point 12 - Page 20, second para, line 7: what is EL?

EL is one of our collaborators Dr. Egil Lien. It is now changed in the text (page 19 line 18).

Point 13 – Page 21, second line from the bottom: 20.000.

The number is now corrected to "20,000 x g" (page 21 line 2).

Point 14 – Page 22, line 2, ripa, Ripa and RIPA were used.

All the words have now been corrected to RIPA.

Point 15 – Page 24: GSE126381, not able to find the data.

The GSE number is correct but the accession is currently private. It is scheduled to be released on July 31st, 2020. It can be seen on the GEO accession (<https://www.ncbi.nlm.nih.gov/geo/query/acc.cgi>) with the following secure token: “bsdkwasvdshzqp”.

Point 16 - References: missing info for 30, 42, and 49.

All the information from the indicated references are now completed.

Point 17 - Figure 1: A, lane labeling was shifted; C, why the cleaved p43 and p18 have difference intensities? Two-tailed nonparametric unpaired t-test, not sure there is a non-parametric t-test? Mann-Whitney?

Lane labeling is corrected (now **Figure 5A**). The antibody we used is from Enzo and recognize p55, p43 and p18. According to our molecular weight ladder the bands correspond to these molecular sizes, as indicated in the immunoblot. It is not clear for us, why the p43 band is weaker. It is now indicated that two-tailed unpaired Mann-Whitney test was performed on **Figure 5** legend.

Point 18 – Figure 2A: The bottom bands should be b-actin?

The bottom bands on Figure 2A (now **Figure 1A**) are beta-actin and are identified on the right side of the blots.

Point 19 – Figure 3B, left: missing labels for the bars.

The labels for the bars are now present in the Figure (now **Supplementary Figure 1B**).

Point 20 - Figure 4C: why higher MFI-Pro-IL-1b in the casp1/11-/- group? 4E top, why Casp8 p55 in non-infected? 4E bottom, no casp8 p43 in one of the casp1/11-/- infected mouse or mis-labeling?

A plausible explanation is that pro-IL-1 β is not being cleaved, and thus accumulate inside the monocytes. However, these differences are not statistically significant (now **Figure 2C**). Sometimes we see a faint band of pro-caspase-8 in purified CD11b cells from spleen of uninfected mice. We believe the process of splenocyte isolation and CD11b cells purification can be a little stressful for the cells, therefore inducing a higher activation background on control samples (now **Figure 2E top**). However, it is clear that there is a major difference of intensity, when comparing infected and non-infected mice. Blot shown on Figure 4E bottom (now **Figure 2E bottom**) is corrected.

Point 21 – Figure 4, nonparametric one-way ANOVA: Kruskal-Wallis?

It is now added on the legend of the Figure that Kruskal-Wallis test was performed (now **Figure 2**).

Point 22 – There are many left/right and top/bottom panels in the figures. It is better to simply label them with a, b, c, sequentially.

As requested, most of the panels are now labeled with different letters.

Reviewer 2:

1. ***“How is caspase 8 driving the IL1b release if it is independent of gasdermin D and RIP3? These observations suggest the working hypothesis of pore formation being required for IL1b release may not be correct, another pore is being formed to release the IL1b or caspase 8 is upregulating another IL1β processing mechanism (this seems unlikely however).”***

We understand the **Reviewer** critique. To further confirm our hypothesis, we generated the RIP3/Caspase8/GSDM-D triple knockout mice. If the main role of caspase-8 was to cleave GSDM-D, the phenotype of RIP3^{-/-}/caspase8^{-/-} and RIP3^{-/-}/caspase8^{-/-}/GSDM-D^{-/-} mice should be similar. However, we found that like the RIP3^{-/-}/caspase8/1/11^{-/-}, the RIP3^{-/-}/caspase8^{-/-}/GSDM-D^{-/-} become highly resistant to ECM. ***These results are now added to Figure 4.*** Furthermore, in various other parameters we measured the phenotype of caspases-1/11 is similar to the GSDM-D single knockouts. In this model, we do not need to evoke a need of a novel pore to release IL-1β. Instead, we favor the hypothesis that, while somewhat redundant, caspase-8 and caspases-1/11 (and GSDM-D) are acting in distinct pathways that seem to be complementary in mediating ECM (Figure 4). Our data suggest that while caspase-8 is controlling the expression of pro-IL-1β and TNFα, caspases-1/11 and GSDM-D are mediating the pathogenesis of malaria by regulating pyroptosis and IL-1β release.

2. ***“In the paper the authors say that “Unlike caspases-1 and -11, the role of caspase-8 does not seem to be IL-1b specific, since its deficiency also impairs the synthesis of other cytokines, such as TNFa and IL-6” Do the authors think that the effect of caspase 8 is mostly related to priming given that caspase 8 can feed into gene regulatory pathways that modify NfκB?***

Overall, the paper would benefit from clarity on the mechanisms by which caspase 8 is regulating IL1b production, processing and release.”

In **Figure 2C**, we show that expression of pro-IL-1β is impaired in the RIP3^{-/-}/caspase-8^{-/-} mice, favoring our main hypothesis that caspase-8 regulates IL-1β transcription.

Minor points

1. **“Furthermore, in oppose to IFNγ^{-/-} mice 24” should read “ Furthermore, opposite to the IFNγ^{-/-} mice” p24**

This now corrected (page 7 line 21).

2. **Fig 2H does not add much as you would expect GasD^{-/-} to not be able to release IL1B**

Considering the point 1 (main comments) raised by this reviewer we believe that the experiments on **Figure 1J** (previous 2H) with GSDMD^{-/-} mice are rather important.

3. **Fig 3A lysate needs to be moved on the figure as the text is on the blot.**

This is now corrected. (Figure 3A is now **Supplementary Figure 1A**)

Reviewer 3:

General comments:

- 1- *In this study, authors show data on human malaria (*P. vivax*), and two mouse models (*P. chabaudi* and *P. berghei*). Although I do understand the need of mouse models to explore some mechanisms, the purpose of these three sets of experiments and how they are interconnected is not clear. For instance: *P. berghei* normally represents severe malaria, especially cases of cerebral malaria in children caused by *P. falciparum*, *P. chabaudi* has been used to understand immune response to mild malaria and how immunity is raised after multiple infections. Authors include some interesting data on the *P. vivax* caspase mechanism; however, it is difficult to link these data to those obtained from mouse models and understand the meaning of them.*

We now try to better connect the data of malaria patients and the two malaria models used in this study both in the introduction and discussion of the manuscript, as well as adding new data of malaria patients.

As requested, we now include the requested immunoblots for the various inflammatory caspases and GSDM-D with samples of patients infected with *P. falciparum*, the etiological agent that cause cerebral malaria (Figure 5).

- 2- *The title mentions cerebral malaria, but why show data obtained from vivax (which does not represent severe complications in this study) and P. chabaudi?*

As suggested, we now change title to **"Caspase-8 mediates inflammation and disease in rodent malaria"** that more broadly indicate the findings in the different malaria disease and models.

Specific comments:

- 1- **Abstract: Authors state that this study suggests the mechanism... Does this mean that the mechanism was not elucidated?**

We now change the word "suggest" to "indicate".

- 2- **Introduction: Text should be revised. There are a few errors. Also: "...In another member of the family, caspase 8..." which family is this? It is not clear why is it relevant to investigate the role of these caspases in malaria pathogenesis? What is the role of caspase 5?**

We now revise the Introduction to address the **Reviewer's** concerns.

- 3- **Results: I believe patient data should go in the end, to confirm the relevance of data obtained from mouse model. They state that caspases 4, 5 and 8 have been activated in vivax infection. Is this correct? If so, this is not clear. In fact, it seems that the vivax data only shows that caspases are activated, but it is not specific. Maybe the authors could better explore *P. vivax* monocyte activation**

by stimulating ex vivo with parasite products instead of LPS. “The results presented in Figures 4A show that the levels of circulating IL-1 β on RIP3- /-caspase-8-/- or RIP3-/-caspase8/1/11-/- mice” Why not the single Caspase 8 KO?

As suggested by the reviewer the data with human monocytes of malaria patients is now moved to the last figure of the manuscript (**Figure 5**)

Based on the immunoblots the different caspases are cleaved generating their active fragments. Thus, we conclude that the *P. vivax* infection is sufficient to activate caspases-1, 4 and 8, as it is also seen for caspases-1, 11 and 8 in the mouse model.

It is known that deletion of caspase-8 gene is embryonic lethal. Nevertheless, simultaneous deletion of RIP3 rescues survival of caspase-8 knockouts. Hence, we used the double knockout and the single RIP3 knockout as negative control. In our model, RIP3 has no role in pathogenesis of malaria. This is explained on page 8 lines 9-11.

4- Discussion: it is stated: “We found that while caspase-11 has a limited role, caspase-8 is essential for IL-1 β release, mediates hypersensitivity to septic shock and cerebral disease in experimental malaria model”. However, in ECM IL-1 β , this does not seem relevant.

We intended to say that caspase-8 is important for IL-1 β expression in both models and it also mediates hypersensitivity to septic shock as well as ECM in the *Pc* and *PbA* mouse models, respectively.

This is now rephrased in the discussion (page 12 line 13-15), as follows:

“We found that while caspase-11 has a limited role, caspase-8 is essential for IL-1 β release and to mediate hypersensitivity to septic shock and cerebral disease in ECM model”.

Reviewers' Comments:

Reviewer #1:

Remarks to the Author:

The manuscript has been improved, and the authors have responded to my queries. This study provides some interesting and important data/mechanism related to malaria induced inflammation.

Reviewer #2:

Remarks to the Author:

The authors have fully addressed my concerns

Reviewer #3:

Remarks to the Author:

The new version of this MS authors answered all my comments point-to-point appropriately. I believe MS is good enough to be accepted.

“Caspase-8 mediates inflammation and disease in rodent malaria”

NCOMMS-19-05667B

POINT-BY-POINT RESPONSE TO REFEREES

We thank all 3 reviewers for accepting the changes and new data in our manuscript and for their overall contribution.

REVIEWERS' COMMENTS:

Reviewer #1 (Remarks to the Author):

The manuscript has been improved, and the authors have responded to my queries. This study provides some interesting and important data/mechanism related to malaria induced inflammation.

No issues were raised by Reviewer #1.

Reviewer #2 (Remarks to the Author):

The authors have fully addressed my concerns

No issues were raised by Reviewer #2.

Reviewer #3 (Remarks to the Author):

The new version of this MS authors answered all my comments point-to-point appropriately. I believe MS is good enough to be accepted.

No issues were raised by Reviewer #3.